# Towards Efficient Online Exploration for Reinforcement Learning with Human Feedback

## Abstract

Reinforcement learning with human feedback (RLHF), which learns a reward model from human preference data and then optimizes a policy to favor preferred responses, has emerged as a central paradigm for aligning large language models (LLMs) with human preferences. In this paper, we investigate exploration principles for online RLHF, where one seeks to adaptively collect new preference data to refine both the reward model and the policy in a data-efficient manner. By examining existing optimism-based exploration algorithms, we identify a drawback in their sampling protocol: they tend to gather comparisons that fail to reduce the most informative uncertainties in reward differences, and we prove lower bounds showing that such methods can incur linear regret over exponentially long horizons. Motivated by this insight, we propose a new exploration scheme that directs preference queries toward reducing uncertainty in reward differences most relevant to policy improvement. Under a multi-armed bandit model of RLHF, we establish regret bounds of order $T^{(\beta+1)/(\beta+2)}$, where $\beta > 0$ is a hyperparameter that balances reward maximization against mitigating distribution shift. To our knowledge, this is the first online RLHF algorithm with regret scaling polynomially in all model parameters.

## 1 Introduction

Large language models (LLMs) have demonstrated remarkable capabilities across a wide range of natural language tasks, yet aligning their behavior with human preferences remains a central challenge. A widely adopted solution is reinforcement learning with human feedback (RLHF), which fine-tunes a pretrained LLM using human preference data (Christiano et al., 2017; Ziegler et al., 2019; Bai et al., 2022). The standard RLHF pipeline involves three stages: (i) supervised fine-tuning (SFT) on human-written demonstrations to produce a baseline model; (ii) training a reward model from human preference comparisons (Bradley & Terry, 1952); and (iii) optimizing the LLM with reinforcement learning against the learned reward. This framework has been instrumental in the success of instruction-following LLMs such as InstructGPT (Ouyang et al., 2022) and ChatGPT (OpenAI, 2023), enabling models to produce responses that are more helpful, safe, and aligned with human expectations.

Despite this progress, most existing RLHF implementations are offline (Zhao et al., 2023; Rafailov et al., 2024; Azar et al., 2024): the preference data is collected once from static policies, and the reward model is trained on this fixed dataset (Ivison et al., 2023; Zhu et al., 2024; Shi et al., 2025). While effective, offline RLHF has inherent limitations—It cannot adaptively explore the enormous space of natural language, leading to inefficient use of expensive human feedback. In contrast, online RLHF offers a more powerful alternative: the policy iteratively collects new preference data, updates the reward model, and improves itself based on these updates (Guo et al., 2024; Xiong et al., 2023; Chen et al., 2024; Rosset et al., 2024; Dong et al., 2024; Feng et al., 2025). This interactive loop has the potential to greatly improve both alignment quality and sample efficiency. However, realizing this potential requires principled approaches to exploration, i.e., deciding which comparisons to query in order to most effectively reduce uncertainty in reward estimation.

A natural candidate for encouraging and guiding exploration is the principle of optimism (Lai & Robbins, 1985; Lattimore & Szepesvári, 2020), which acts as if the environment is more optimistic than currently estimated, within the limits of statistical uncertainty based on all data that has been observed so far. It is usually implemented by adding an uncertainty-based bonus to reward or value estimates, thereby prioritizing actions whose values are uncertain but potentially high. This has yielded provably efficient algorithms in standard RL (see e.g., Jin et al. (2018); Zanette & Brunskill (2019); Russo & Van Roy (2013); Azar et al. (2017)). However, extending this principle to RLHF introduces new difficulties, where feedback comes not as a single reward but as a difference between rewards of two actions. The key challenge is to determine the action pairs with the large uncertainties most relevant to policy improvement. A few recent works achieved important progress towards designing sample-efficient online RLHF algorithms based on the optimism principle (Cen et al., 2025; Zhang et al., 2025; Xie et al., 2025). However the existing theoretical guarantees still exhibit exponential dependency on certain model parameters, which potentially leads to inefficient exploration.

With this context, this paper makes contribution towards designing efficient online exploration schemes for RLHF with provable guarantees. By analyzing the existing algorithms in the seminal works (Cen et al., 2025; Xie et al., 2025; Zhang et al., 2025), we discuss their inadequacy in exploring the action pairs with the large uncertainties most relevant to policy improvement, and construct lower bounds to show that the exponential dependency on certain parameters is unavoidable in their regret. Based on these insights, we propose a new exploration scheme for RLHF that adopts a different sampling protocol, and establish a regret bound that depends polynomially on all model parameters.

## 2 MODEL SET-UP

**Preliminaries.** In RLHF, the prompt space $\mathcal{X}$ refers to the collection of all possible inputs or queries that a user might provide to the model. The answer (or action) space $\mathcal{A}$ is the set of all possible outputs the model can generate in reply to a given prompt. A language model is a policy $\pi : \mathcal{X} \to \Delta(\mathcal{A})$ that defines a probability distribution $\pi(\cdot \,|\, x)$ over $\mathcal{A}$ conditioned on a prompt $x \in \mathcal{X}$, specifying how likely the model is to produce each potential response. The pipeline of RLHF starts with supervised fine-tuning (SFT), where a reference policy $\pi_{\mathsf{ref}} : \mathcal{X} \to \Delta(\mathcal{A})$ is obtained by fine-tuning a pre-trained LLM on a dataset of prompts paired with high-quality answers written by humans. SVT provides an initialization that stabilizes and improves the effectiveness of the subsequent training stages that aligns the LLM with human preferences.

**Reward modeling.** To translate human preferences into a trainable objective, one need to model how an oracle (e.g., a human annotator) rank two answers $a_1$ and $a_2$ given prompt $x$. Following a line of prior works (e.g., Cen et al. (2025); Xie et al. (2025); Zhang et al. (2025)), we assume that preferences follow the Bradley-Terry model (Bradley & Terry, 1952)

$$\mathbb{P}(a_1 \succ a_2 \,|\, x) = \frac{\exp(r^\star(x, a_1))}{\exp(r^\star(x, a_1)) + \exp(r^\star(x, a_2))} = \sigma\left(r^\star(x, a_1) - r^\star(x, a_2)\right). \tag{2.1}$$

Here $r^\star : \mathcal{X} \times \mathcal{A} \to \mathbb{R}$ is an underlying reward function of an answer given a prompt, $a_1 \succ a_2$ means the answer $a_1$ is preferred compared to $a_2$, and $\sigma(x) = (1 + e^{-x})^{-1}$ is the sigmoid function. We also define a policy $\pi_{\mathsf{HF}}$ to characterize human preference:

$$\pi_{\mathsf{HF}}(a \,|\, x) = \frac{\exp(r^\star(x, a))}{\sum_{a' \in \mathcal{A}} \exp(r^\star(x, a'))}.$$

The reward function is unknown and can be learned from e.g., an offline dataset $\mathcal{D} = \{(x^i, a_+^i, a_-^i)\}$ comprised of independent preference data samples using maximum likelihood estimation (MLE):

$$\arg\max_r \ell(r, \mathcal{D}) \quad \text{where} \quad \ell(r, \mathcal{D}) := \sum_{\mathcal{D}} \log \sigma\left(r(x^i, a_+^i) - r(x^i, a_-^i)\right), \tag{2.2}$$

where a preference data sample denoted by $(x, a_+, a_-)$ means that $a_+ \succ a_-$ given prompt $x$.

**RL fine-tuning.** Given a reward model $r$, we seek to fine-tune the policy $\pi$ to balance reward maximization with maintaining similarity to the original model $\pi_{\text{ref}}$ from the SFT stage. Towards this, we define the KL-regularized reward objective

$$J(\pi, r; \pi_{\text{cal}}) \coloneqq \mathbb{E}_{x \sim \rho}\big[\mathbb{E}_{a \sim \pi(\cdot \,|\, x)}[r(x,a)] - \mathbb{E}_{a \sim \pi_{\text{cal}}(\cdot \,|\, x)}[r(x,a)] - \beta \mathsf{KL}\big(\pi(\cdot \,|\, x) \,\|\, \pi_{\text{ref}}(\cdot \,|\, x)\big)\big]. \tag{2.3}$$

Here $\rho$ is the prompt distribution, and $\beta > 0$ is the regularization parameter reflecting the strength of the KL regularization. In practice, $\beta$ is typically chosen to be small; for instance, in InstructGPT (Ouyang et al., 2022) the optimal value is reported to be around $0.01$ and $0.02$. This objective function includes a calibration policy $\pi_{\text{cal}}$ to eliminate the shift ambiguity of the reward function, as two reward functions $r(x,a)$ and $r(x,a) + c(x)$ lead to the same preference model (2.1). Given any reward function $r$, the optimal policy $\pi_r \coloneqq \arg\max_\pi J(\pi, r; \pi_{\text{cal}})$ admits a closed-form expression (Rafailov et al., 2024)

$$\pi_r(a \,|\, x) = \frac{\pi_{\text{ref}}(a \,|\, x) \exp(r(x,a)/\beta)}{Z_r(x)} \tag{2.4}$$

where $Z_r(x) = \sum_a \pi_{\text{ref}}(a|x) \exp(r(x,a)/\beta)$ is the normalizing factor. Notice that the selection of $\pi_{\text{cal}}$ does not affect the optimal policy $\pi_r$ given the reward function $r$. Our target is the optimal policy $\pi^\star$ that maximizes the objective (2.3) under the true reward function $r = r^\star$, namely

$$\pi^\star \coloneqq \arg\max_\pi J(\pi, r^\star; \pi_{\text{cal}}). \tag{2.5}$$

**Offline RLHF.** The above framework leads to offline RLHF methods that relies on the preference dataset $\mathcal{D}$ for training. Initial approaches (Christiano et al., 2017; Ouyang et al., 2022) first estimate a reward function $\widehat{r}$ based on the preference dataset $\mathcal{D}$ using MLE, then optimize the KL-regularized objective (2.3) with respect to $\widehat{r}$. Another approach introduced by Rafailov et al. (2024) condensed these two steps into one single step, known as direct preference optimization (DPO), which optimizes

$$\max_\pi \sum_{\mathcal{D}} \log \sigma\bigg(\beta\Big(\log \frac{\pi(y_+^i \,|\, x)}{\pi_{\text{ref}}(y_+^i \,|\, x)} - \log \frac{\pi(y_-^i \,|\, x)}{\pi_{\text{ref}}(y_-^i \,|\, x)}\Big)\bigg).$$

The above objective avoids explicitly estimating the reward function, which can be obtained by expressing the reward function $r$ in the MLE formulation (2.2) with the associated optimal policy $\pi_r$ using the closed-form expression (2.4). However, as discussed in e.g., Xie et al. (2025); Zhang et al. (2025), the efficiency of offline RLHF is limited by the coverage of the offline dataset $\mathcal{D}$, and online exploration with active data collection is necessary to achieve sample efficiency.

**Online RLHF.** We consider reward learning and policy learning iteratively, where in the $t$-th iteration we use the current policy $\pi^{(t)}$, obtained from previous iterations, to sample new data and subsequently update both the reward estimate and the policy. This setup enables online exploration in RLHF by refining the reward model and policy in tandem as new preference data is collected. We aim to minimize the regret

$$\mathcal{R}(T) \coloneqq \sum_{t=1}^{T} \big[J(\pi^\star; r^\star, \pi_{\text{cal}}) - J(\pi^{(t)}; r^\star, \pi_{\text{cal}})\big]. \tag{2.6}$$

It is worth mentioning that the choice of $\pi_{\text{cal}}$ does not affect the regret. We define the following function $J^\star$ that measures the optimal objective value for a given reward $r$:

$$J^\star(r; \pi_{\text{cal}}) \coloneqq \max_\pi J(\pi, r; \pi_{\text{cal}}) = J(\pi_r, r; \pi_{\text{cal}}). \tag{2.7}$$

This function plays an important role in the exploration algorithms.

## 3  RLHF with online exploration

Three recent algorithms for online RLHF are most closely related to this work: VPO (Cen et al., 2025), XPO (Xie et al., 2025), and SELM (Zhang et al., 2025). In this section, we first analyze and discuss these approaches, and then introduce our proposed exploration scheme.

## 3.1 INADEQUACY OF EXISTING APPROACHES

We begin by reviewing the procedure and intuition behind VPO (Cen et al., 2025). Fix a calibration policy $\pi_{\mathsf{cal}}$ and an initial policy $\pi^{(1)}$. For $t = 1, 2, \ldots, T$, the $t$-th iteration of VPO consists of the following steps:

1. Sample a prompt $x^t \sim \rho$ and two answers $a_1^t, a_2^t \sim \pi^{(t)}(\cdot \mid x^t)$. Query the preference oracle to obtain pairwise comparison $a_+^t \succ a_-^t$. Update the preference dataset $\mathcal{D}^{(t)} = \mathcal{D}^{(t-1)} \cup \{(x^t, a_+^t, a_-^t)\}$.

2. Update the reward model $r^{(t+1)}$ and the policy $\pi^{(t+1)}$ using the updated preference dataset $\mathcal{D}^{(t)}$:

$$r^{(t+1)} = \underset{r: \mathcal{X} \times \mathcal{A} \to [0, r_{\max}]}{\arg\max} \ell(r, \mathcal{D}^{(t)}) + \alpha J^\star(r; \pi_{\mathsf{cal}}), \tag{3.1a}$$

$$\pi^{(t+1)} = \underset{\pi}{\arg\max} \, J(\pi, r^{(t+1)}; \pi_{\mathsf{cal}}), \tag{3.1b}$$

where $\alpha > 0$ is a regularization parameter, and step (3.1b) admits closed-form solution (2.4).

To illustrate the rationale behind VPO, consider the bandit case with no prompt. Step (3.1a) applies the optimism principle, encouraging exploration based on the uncertainty in estimating the reward difference between each action $a$ and the calibration policy $\pi_{\mathsf{cal}}$. Formally, it can be viewed as the Lagrangian form of the constrained optimization problem

$$\max_{r, \pi} \mathbb{E}_{a \sim \pi}[r(a)] - \mathbb{E}_{a \sim \pi_{\mathsf{cal}}}[r(a)] - \beta \mathsf{KL}(\pi \parallel \pi_{\mathsf{ref}}) \quad \text{s.t.} \quad \ell(r, \mathcal{D}^{(t)}) \geq \max_r \ell(r, \mathcal{D}^{(t)}) - B$$

for some $B > 0$. After the change of variable $r'(a) = r(a) - \mathbb{E}_{a \sim \pi_{\mathsf{cal}}}[r(a)]$, this becomes

$$\max_{r', \pi} \mathbb{E}_{a \sim \pi}[r'(a)] - \beta \mathsf{KL}(\pi \parallel \pi_{\mathsf{ref}}) \quad \text{s.t.} \quad \ell(r', \mathcal{D}^{(t)}) \geq \max_{r'} \ell(r', \mathcal{D}^{(t)}) - B, \quad \mathbb{E}_{a \sim \pi_{\mathsf{cal}}}[r'(a)] = 0.$$

Here, the constraint set can be interpreted as a confidence region reflecting the uncertainty in estimating each $r'(a)$ from $\mathcal{D}^{(t)}$. Consequently, the updated policy $\pi^{(t+1)}$ depends both on the true reward gap $r(a) - \mathbb{E}_{a \sim \pi_{\mathsf{cal}}}[r(a)]$ and on the uncertainty in estimating this gap for each action $a \in \mathcal{A}$.

For intuition, suppose $\pi_{\mathsf{cal}} = \mathbb{1}_{a_0}$ for some $a_0 \in \mathcal{A}$, and assume that the true reward gaps are small. In this case, $\pi^{(t+1)}$ favors actions with higher estimation uncertainty relative to $a_0$, i.e., those $a$ where the estimate of $r(a) - r(a_0)$ is most uncertain. However, comparing two actions $a_1, a_2 \sim \pi^{(t+1)}$ reduces the uncertainty between them, rather than the (potentially larger) uncertainty relative to $a_0$. This misalignment can lead to inefficient exploration, as illustrated in the following example.

**Example 1.** *Consider the bandit setting with three actions $\mathcal{A} = \{a_0, a_1, a_2\}$, where the true rewards are $r^\star(a_0) = 1$ and $r^\star(a_1) = r^\star(a_2) = 0$. Let the reference policy $\pi_{\mathsf{ref}}$ be uniform over $\mathcal{A}$, and the calibration policy be $\pi_{\mathsf{cal}}(a_1) = \pi_{\mathsf{cal}}(a_2) = p$ and $\pi_{\mathsf{cal}}(a_0) = 1 - 2p$ for some $0 \leq p < 1/4$.*

The following proposition shows that VPO may fail to explore efficiently in this setting The proof can be found in Appendix A.

**Proposition 1.** *Consider the setup in Example 1. Let the initial policy $\pi^{(1)}$ of VPO be the uniform distribution over $\mathcal{A}$. Assume that $r_{\max}/\beta \geq 3$. For any $\alpha > 0$, with probability at least $4/(9e)$, we have*

$$J(\pi^\star, r^\star; \pi_{\mathsf{cal}}) - J(\pi^{(t)}, r^\star; \pi_{\mathsf{cal}}) \geq \frac{1}{2}$$

*holds for any $1 < t \leq \exp(r_{\max}/\beta)/2$.*

Let's discuss the idea behind Proposition 1 with $\pi_{\mathsf{cal}} = \mathbb{1}_{a_0}$. If the calibration action $a_0$ is not visited during the first $t$ iterations, then $\pi^{(t+1)}$ will continue to favor $a_1$ and $a_2$, since both gaps $r(a_1) - r(a_0)$ and $r(a_2) - r(a_0)$ remain highly uncertain. In particular, we establish that $\pi^{(t+1)}(a_0) \leq \exp(-r_{\max}/\beta)$, which is exponentially small, implying that $a_0$ is unlikely to be sampled in iteration $t + 1$. As a result, with constant probability, $a_0$ will not be sampled within the first $O(\exp(r_{\max}/\beta))$ iterations, and the resulting highly suboptimal policy incurs linear regret over an exponentially long horizon. This example highlights an algorithmic drawback: although VPO acknowledges uncertainty in the reward gaps between $a_1$ and $a_0$ (and between $a_2$ and $a_0$), it continues to encourage sampling $a_1$ and $a_2$, leading primarily to comparisons between them that fail to reduce their uncertainty relative to $a_0$.

---

**Algorithm 1:** Uncertainty-based RLHF exploration.

1 **Input:** initial policies $\pi^{(0)}, \pi^{(1)}$, regularizaton parameters $\{\alpha_t\}_{t \geq 1}$.

2 **for** $t = 1$ **to** $T$ **do**

3      Sample a prompt $x^t \sim \rho$ and two answers $a_1^t \sim \pi^{(t-1)}(\cdot \,|\, x^t)$, $a_2^t \sim \pi^{(t)}(\cdot \,|\, x^t)$.

4      Query the preference oracle to obtain pairwise comparison $a_+^t \succ a_-^t$ and update the preference dataset $\mathcal{D}^{(t)} = \mathcal{D}^{(t-1)} \cup \{(x^t, a_+^t, a_-^t)\}$.

5      Update the reward model $r^{(t+1)}$ and the policy $\pi^{(t+1)}$ using $\mathcal{D}^{(t)}$:

$$r^{(t+1)} = \underset{r:\mathcal{X} \times \mathcal{A} \to [0, r_{\max}]}{\arg \max} \ell(r, \mathcal{D}^{(t)}) + \alpha_t J^\star(r; \pi^{(t)}), \tag{3.2a}$$

$$\pi^{(t+1)} = \underset{\pi}{\arg \max} \, J(\pi, r^{(t+1)}; \pi^{(t)}). \tag{3.2b}$$

     where the policy update (3.2b) admits closed-form solution (2.4).

6 **Output:** $\{\pi^{(t)} : 1 \leq t \leq T\}$

---

### 3.2 OUR APPROACH: EXPLORATION BASED ON UNCERTAINTY

A natural modification to address the issue above is to change the sampling scheme so that $a_1^t \sim \pi^{(t)}$ and $a_2^t \sim \pi_{\mathsf{cal}}$. The intuition is that $\pi^{(t)}$ encourages to explore actions with higher estimation uncertainty relative to the actions favored by the calibration policy $\pi_{\mathsf{cal}}$. To effectively reduce this uncertainty, it is sensible to compare one action drawn from $\pi^{(t)}$ with another drawn from $\pi_{\mathsf{cal}}$. Indeed, the XPO and SELM algorithms (Xie et al., 2025; Zhang et al., 2025) can be viewed as taking $\pi_{\mathsf{cal}} = \pi_{\mathsf{ref}}$.

However, if the fixed calibration policy $\pi_{\mathsf{cal}}$ is highly suboptimal for reward maximization (for example, if it concentrates on a few low-reward actions), then the comparison will almost always favor $a_1^t \sim \pi^{(t)}$ against $a_2^t \sim \pi_{\mathsf{cal}}$, yielding little useful information. This issue is illustrated in the following example.

**Example 2.** *Consider the bandit setting with three actions $\mathcal{A} = \{a_0, a_1, a_2\}$, where the true rewards are $r^\star(a_0) = 0$, $r^\star(a_1) = r_{\max}$ and $r^\star(a_2) = r_{\max} - 2$. Let the reference policy be $\pi_{\mathsf{ref}}(a_0) = 1 - 2/\kappa$, $\pi_{\mathsf{ref}}(a_1) = \pi_{\mathsf{ref}}(a_2) = 1/\kappa$ for any $\kappa \geq 4$.*

The following result shows that, when $\kappa$ is large (as we will see in Assumption 1, this corresponds to the case where the reference policy deviates from human preference), this modified sampling schemes can lead to inefficient exploration in this setting. The proof is deferred to Appendix B.

**Proposition 2.** *Consider the setup in Example 2. Assume that $\beta \leq 1$ and $\kappa \leq \exp(r_{\max}/\beta)$. For any initial policy $\pi^{(1)}$ and any $\alpha > 0$, with probability at least $1/64$, the modified exploration scheme which samples $a_1^t \sim \pi^{(t)}$ and $a_2^t \sim \pi_{\mathsf{ref}}$ satisfies*

$$J(\pi^\star, r^\star; \pi_{\mathsf{ref}}) - J(\pi^{(t)}, r^\star; \pi_{\mathsf{ref}}) \geq 0.01$$

*for any $1 < t \leq \min\{\kappa, \exp(r_{\max})/2\}$.*

This lower bound suggests that relying on a fixed calibration policy can lead to inefficient exploration over an exponentially long horizon. We will come back to this example in Section 4 after presenting our algorithm and theoretical guarantees. This observation motivates us to update the calibration policy in each iteration adaptively.

**Uncertainty-based exploration.** We propose an exploration scheme where the calibration policy evolves with the iterations. In the $t$-th iteration, instead of a fixed $\pi_{\mathsf{cal}}$, we use $\pi^{(t)}$ as the calibration policy when optimizing $r^{(t+1)}$ and $\pi^{(t+1)}$:

$$r^{(t+1)} = \underset{r:\mathcal{X} \times \mathcal{A} \to [0, r_{\max}]}{\arg \max} \ell(r, \mathcal{D}^{(t)}) + \alpha_t J^\star(r; \pi^{(t)}),$$

$$\pi^{(t+1)} = \underset{\pi}{\arg \max} \, J(\pi, r^{(t+1)}; \pi^{(t)}).$$

The key advantage is that $\pi^{(t)}$ improves over time, guiding exploration away from uninformative comparisons. Since $\pi^{(t)}$ emphasizes actions with higher uncertainty relative to $\pi^{(t-1)}$, it is natural to compare $a_1^t \sim \pi^{(t-1)}$ and $a_2^t \sim \pi^{(t)}$. This yields preference data that more directly reduces uncertainty, leading to more efficient exploration. Our full exploration scheme is summarized in Algorithm 1.

## 4 THEORETICAL RESULTS

We establish theoretical guarantees for Algorithm 1 under the multi-armed bandit setting (i.e., $\mathcal{X} = \varnothing$) with $A = |\mathcal{A}|$. We begin with a general regret bound, whose proof is deferred to Section 5.

**Theorem 1.** *Let $\alpha_t > A \log T$ be non-decreasing in $t$. There exists a universal constant $C > 0$ such that, with probability at least $1 - O(T^{-10})$, the cumulative regret of running Algorithm 1 for $T$ iterations satisfies*

$$\mathcal{R}(T) \leq C r_{\mathsf{max}} A^2 \sqrt{T \log T} + C \sum_{t=1}^{T} \frac{A r_{\mathsf{max}} \log T}{\alpha_t} + C A^2 \alpha_T r_{\mathsf{max}}^2 \tag{4.1}$$

$$+ C(r_{\mathsf{max}} + \log T) \sum_{r^\star(a_+) \geq r^\star(a_-)} \min \left\{ \frac{\pi_{\mathsf{HF}}(a_+)}{\pi_{\mathsf{HF}}(a_-)} \alpha_T r_{\mathsf{max}}, \left( T \frac{\pi_{\mathsf{ref}}(a_-)}{\pi_{\mathsf{ref}}(a_+)} \right)^{\frac{\beta}{\beta+1}} \alpha_T^{\frac{1}{\beta+1}} r_{\mathsf{max}}^{\frac{1}{\beta+1}} \right\}.$$

We now discuss the implications of Theorem 1. When $\beta = 0$, which corresponds to the case where only reward maximization matters, the regret bound (4.1) simplifies to

$$\mathcal{R}(T) = \widetilde{O}\big( (A^{3/2} r_{\mathrm{max}}^{3/2} + A^2 r_{\mathrm{max}}) \sqrt{T} \big) \qquad \text{when} \qquad \alpha_t \asymp A \log T + \sqrt{\frac{t}{A r_{\mathrm{max}}}}.$$

When $\beta > 0$, the performance of the exploration algorithm becomes more intricate due to the trade-off between reward maximization and similarity to the reference policy. To interpret the general regret bound in this regime, we introduce the following assumption to capture the interaction between human preference $\pi_{\mathsf{HF}}$ and the reference policy $\pi_{\mathsf{ref}}$.

**Assumption 1.** *There exists $\kappa, \tau \geq 1$ such that, for any action pair $(a_+, a_-)$,*

$$\frac{\pi_{\mathsf{HF}}(a_+)}{\pi_{\mathsf{HF}}(a_-)} \geq \tau \quad \implies \quad \frac{\pi_{\mathsf{ref}}(a_+)}{\pi_{\mathsf{ref}}(a_-)} \geq \kappa^{-1}.$$

Intuitively, Assumption 1 requires that whenever $a_+$ is substantially more preferred than $a_-$ under human preference, the reference policy does not assign disproportionately higher weight to $a_-$ than to $a_+$. This is reasonable, since $\pi_{\mathsf{ref}}$ is obtained from the SFT step, where a pretrained LLM is fine-tuned on human demonstrations already broadly aligned with preference. The quantities $\kappa$ and $\tau$ capture the degree of alignment between $\pi_{\mathsf{ref}}$ and $\pi_{\mathsf{HF}}$, and their size reflects the influence of the reference policy on RLHF. We note that the illustrative Example 1 satisfies Assumption 1 with $\kappa, \tau = O(1)$, and the parameter $\kappa$ in Example 2 is consistent with the $\kappa$ here. Under this assumption, we obtain the following simplified regret bound, whose proof is deferred to Appendix D.

**Proposition 3.** *Suppose that Assumption 1 holds. Let*

$$\alpha_t = A \log T + t^{\frac{1}{\beta+2}} \left( \frac{r_{\mathrm{max}}}{\kappa} \right)^{\frac{\beta}{\beta+2}} \left( \frac{\log T}{A(r_{\mathrm{max}} + \log T)} \right)^{\frac{\beta+1}{\beta+2}}.$$

*Then with probability at least $1 - O(T^{-10})$, we have*

$$\mathcal{R}(T) \lesssim (\tau + \kappa^\beta T^{\frac{\beta+1}{\beta+2}}) \, \mathsf{poly}(A, r_{\mathrm{max}}, \log T),$$

*where the degree of the polynomial factor does not depend on $\beta$.*

***Remark* 1.** When $\kappa$ is large, namely the reference policy deviates significantly from the human preference, it is natural to choose a small KL regularization parameter $\beta$ to reduce the influence of the reference policy. In this regime, Algorithm 1 remains robust, since the regret bound scales only with $\kappa^\beta$. By contrast, the lower bound in Proposition 2 suggests that the sampling protocols in prior works (Xie et al., 2025; Zhang et al., 2025) would incur regret at least linear in $\kappa$. This demonstrates that our strategy accommodates scenarios with small $\beta$, where the reference policy is poorly aligned with human preference.

**Remark** 2. In Appendix E, we present an alternative assumption linking human preference and the reference policy, together with the corresponding regret guarantee.

Proposition 3 establishes a regret bound of order $O(T^{\frac{\beta+1}{\beta+2}})$, with only polynomial dependence on the other parameters. This stands in sharp contrast to prior works (Cen et al., 2025; Xie et al., 2025; Zhang et al., 2025), which achieved the more standard $O(\sqrt{T})$ regret but at the cost of exponential dependence on terms such as $r_{\max}/\beta$. We conjecture that, for RLHF, eliminating exponential dependence inevitably requires a slower rate in $T$, with the exponent governed by $\beta$. This trade-off is intuitive: online exploration primarily serves to learn human preference, and as the regularization parameter $\beta$ increases, greater emphasis is placed on preserving similarity to the reference measure. This constraint naturally slows convergence.

## 5   PROOF OF THEOREM 1

### 5.1   STEP 1: REGRET DECOMPOSITION

In view of the optimality of $r^{(t)}$ (cf. equation (3.2a)), we have

$$\ell(r^{(t)}, \mathcal{D}^{(t-1)}) + \alpha_t J^\star(r^{(t)}; \pi^{(t-1)}) \geq \ell(r^\star, \mathcal{D}^{(t-1)}) + \alpha_t J^\star(r^\star; \pi^{(t-1)}).$$

Rearrange terms to get

$$\frac{1}{\alpha_t}\big[\ell(r^{(t)}, \mathcal{D}^{(t-1)}) - \ell(r^\star, \mathcal{D}^{(t-1)})\big] \geq J^\star(r^\star; \pi^{(t-1)}) - J^\star(r^{(t)}; \pi^{(t-1)})$$

$$\stackrel{\text{(i)}}{=} \max_\pi J(\pi, r^\star; \pi^{(t-1)}) - \max_\pi J(\pi, r^{(t)}; \pi^{(t-1)})$$

$$\stackrel{\text{(ii)}}{\geq} J(\pi^\star, r^\star; \pi^{(t-1)}) - J(\pi^{(t)}, r^{(t)}; \pi^{(t-1)}). \tag{5.1}$$

Here step (i) follows from the definition of $J^\star$ (cf. equation (2.7)), while step (ii) follows from the optimality of $\pi^{(t)}$ (cf. equation (3.2b)). This allows us to reach the following decomposition:

$$\text{Regret}_t \coloneqq J(\pi^\star, r^\star; \pi^{(t-1)}) - J(\pi^{(t)}, r^\star; \pi^{(t-1)})$$

$$\leq \underbrace{\alpha_t^{-1}\big[\ell(r^{(t)}, \mathcal{D}^{(t)}) - \ell(r^\star, \mathcal{D}^{(t)})\big]}_{=:\theta_t} + \underbrace{J(\pi^{(t)}, r^{(t)}; \pi^{(t-1)}) - J(\pi^{(t)}, r^\star; \pi^{(t-1)})}_{=:\gamma_t}. \tag{5.2}$$

In view of the definition of $J$ (cf. equation (2.3)), we can further decompose

$$\gamma_t = \mathbb{E}_{a\sim\pi^{(t)}}[r^{(t)}(a)] - \mathbb{E}_{a\sim\pi^{(t-1)}}[r^{(t)}(a)] - \mathbb{E}_{a\sim\pi^{(t)}}[r^\star(a)] + \mathbb{E}_{a\sim\pi^{(t-1)}}[r^\star(a)]$$

$$= r^{(t)}(a_2^t) - r^{(t)}(a_1^t) - r^\star(a_2^t) + r^\star(a_1^t) + \xi_t$$

where $\xi_t$ is the martingale difference sequence

$$\xi_t = \mathbb{E}_{a\sim\pi^{(t)}}[r^{(t)}(a)] - r^{(t)}(a_2^t) - \mathbb{E}_{a\sim\pi^{(t-1)}}[r^{(t)}(a)] + r^{(t)}(a_1^t)$$

$$- \mathbb{E}_{a\sim\pi^{(t)}}[r^\star(a)] + r^\star(a_2^t) + \mathbb{E}_{a\sim\pi^{(t-1)}}[r^\star(a)] - r^\star(a_1^t).$$

Therefore we have

$$\text{Regret} = \sum_{t=1}^T \text{Regret}_t \leq \underbrace{\sum_{t=1}^T \theta_t}_{=:\theta} + \underbrace{\sum_{t=1}^T \xi_t}_{=:\xi} + \underbrace{\sum_{t=1}^T |r^{(t)}(a_2^t) - r^{(t)}(a_1^t) - r^\star(a_2^t) + r^\star(a_1^t)|}_{=:\zeta}. \tag{5.3}$$

It is straightforward to bound the second term $\xi$. Notice that $|\xi_t| \leq 8r_{\max}$ holds deterministically for any $1 \leq t \leq T$. By the Azuma-Hoeffding inequality, with probability exceeding $1 - O(T^{-10})$ we have

$$\xi = \sum_{t=1}^T \xi_t \leq C_1 r_{\max}\sqrt{T\log T} \tag{5.4}$$

for some universal constant $C_1 > 0$. In what follows, we bound the other two terms $\theta$ and $\zeta$.

## 5.2 STEP 2: BOUNDING LIKELIHOOD RATIOS

To bound $\theta$, we need to analyze the regularized MLE. Notice that

$$\theta_t = \frac{\ell(r^{(t)}, \mathcal{D}^{(t)}) - \ell(r^\star, \mathcal{D}^{(t)})}{\alpha_t} = \alpha_t^{-1} \sum_{i=1}^{t} \log \frac{\sigma(r^{(t)}(x^i, a_+^i) - r^{(t)}(x^i, a_-^i))}{\sigma(r^\star(x^i, a_+^i) - r^\star(x^i, a_-^i))}.$$

The following lemma is crucial for the subsequent analysis. The proof can be found in Appendix C.1.

**Lemma 1.** *For any given reward function $r : \mathcal{A} \to [\pm r_{\max}]$ and any $1 \le t \le T$, define*

$$\Delta_t(r) := \sum_{i=1}^{t} \log \frac{\sigma(r^\star(a_+^i) - r^\star(a_-^i))}{\sigma(r(a_+^i) - r(a_-^i))} - \sum_{i=1}^{t} \mathsf{KL}\big(\sigma(r^\star(a_1^i) - r^\star(a_2^i)) \,\|\, \sigma(r(a_1^i) - r(a_2^i))\big).$$

*There exists some universal constant $C_2 > 1$ such that for any fixed $r$, with probability at least $1 - \delta$,*

$$|\Delta_t(r)| \le C_2 \sqrt{\sum_{i=1}^{t} r_{\max} \mathsf{KL}\big(\sigma(r^\star(a_1^i) - r^\star(a_2^i)) \,\|\, \sigma(r(a_1^i) - r(a_2^i))\big) \log \frac{\log T}{\delta}} + C_2 r_{\max} \log \frac{\log t}{\delta}.$$

Equipped with the concentration bounds in Lemma 1, we can use the standard covering argument to derive an uniform upper bound, whose proof is deferred to Appendix C.2.

**Lemma 2.** *There exists some universal constant $C_3 > 0$ such that with probability exceeding $1 - O(T^{-9})$,*

$$\ell(r, \mathcal{D}^{(t)}) - \ell(r^\star, \mathcal{D}^{(t)}) \le -\frac{1}{2} \sum_{i=1}^{t} \mathsf{KL}\big(\sigma(r^\star(a_1^i) - r^\star(a_2^i)) \,\|\, \sigma(r(a_1^i) - r(a_2^i))\big) + C_3 A r_{\max} \log T$$

*holds for any $r : \mathcal{A} \to [\pm r_{\max}]$ and $1 \le t \le T$.*

As an immediate consequence of Lemma 2, with probability exceeding $1 - O(T^{-9})$,

$$\ell(r^{(t)}, \mathcal{D}^{(t)}) - \ell(r^\star, \mathcal{D}^{(t)}) \le C_3 A r_{\max} \log T$$

holds for any $1 \le t \le T$. Therefore

$$\theta = \sum_{t=1}^{T} \theta_t = \sum_{t=1}^{T} \frac{\ell(r^{(t)}, \mathcal{D}^{(t)}) - \ell(r^\star, \mathcal{D}^{(t)})}{\alpha_t} \le \sum_{t=1}^{T} \frac{C_3 A r_{\max} \log T}{\alpha_t}. \tag{5.5}$$

## 5.3 STEP 3: BOUNDING REWARD ERRORS

We first notice that

$$\alpha_t^{-1}\big[\ell(r^{(t)}, \mathcal{D}^{(t-1)}) - \ell(r^\star, \mathcal{D}^{(t-1)})\big] \overset{\text{(i)}}{\ge} \max_\pi J(\pi, r^\star; \pi^{(t-1)}) - \max_\pi J(\pi, r^{(t)}; \pi^{(t-1)})$$

$$\overset{\text{(ii)}}{\ge} J(\pi^{(t)}, r^\star; \pi^{(t-1)}) - J(\pi^{(t)}, r^{(t)}; \pi^{(t-1)})$$

$$\overset{\text{(iii)}}{=} \mathbb{E}_{a \sim \pi^{(t)}}[r^{(t)}(a) - r^\star(a)] - \mathbb{E}_{a \sim \pi^{(t-1)}}[r^{(t)}(a) - r^\star(a)]$$

$$\ge -4 r_{\max}. \tag{5.6}$$

Here step (i) is an intermediate step of (5.1); step (ii) follows from the optimality of $\pi^{(t)}$ (cf. (3.2b)); step (iii) follows from the definition of $J$ (cf. (2.3)). This combined with Lemma 2 implies that

$$\sum_{i=1}^{t} \mathsf{KL}\big(\sigma(r^\star(a_1^i) - r^\star(a_2^i)) \,\|\, \sigma(r^{(t)}(a_1^i) - r^{(t)}(a_2^i))\big) \tag{5.7}$$

$$\le -2\big[\ell(r^{(t)}, \mathcal{D}^{(t)}) - \ell(r^\star, \mathcal{D}^{(t)})\big] + 2 C_3 A r_{\max} \log T \le C_4 \alpha_t r_{\max},$$

as long as $\alpha_t \ge A \log T$ and $C_4 \ge 8 + 2 C_3$. This implies that for any $t \in [T]$ and any action pair $(a_+, a_-)$,

$$\mathsf{KL}\big(\sigma(r^\star(a_+) - r^\star(a_-)) \| \sigma(r^{(t)}(a_+) - r^{(t)}(a_-))\big) \le \frac{C_4 \alpha_t r_{\max}}{N_t(a_+, a_-)}, \tag{5.8}$$

where $N_t(a_+, a_-)$ is the number of comparison for $(a_+, a_-)$ up to time $t$. This motivates us to decompose $\zeta$ according to whether $N_t(a_+, a_-) \gg \alpha_t r_{\max}$: let $\tau := 100 C_4 \alpha_T r_{\max}$ and denote by $t_n(a_+, a_-)$ the time of the $n$-th comparison for $(a_+, a_-)$, we have

$$\zeta \leq 2\tau A^2 r_{\max} + \underbrace{\sum_{r^\star(a_+) \geq r^\star(a_-)} \sum_{n=\tau}^{N_T(a_+, a_-)} \left| r^{(t_n)}(a_+) - r^{(t_n)}(a_-) - r^\star(a_+) + r^\star(a_-) \right|}_{=: \zeta(a_+, a_-)},$$

where we denote by $t_n(a_+, a_-)$ the time of the $n$-th comparison for $(a_+, a_-)$, and the first summation is taken over all action pairs $(a_+, a_-)$ satisfying $r^\star(a_+) \geq r^\star(a_-)$. To bound each $\zeta(a_+, a_-)$, we need the following technical lemma. The proof can be found in Appendix C.3.

**Lemma 3.** *Consider any action pair $(a_+, a_-)$ and time $t_0$ such that $N_{t_0}(a_+, a_-) \geq \tau$. There exists universal constant $C_5 > 0$ such that, for any $t_0 \leq t_1 < t_2 \leq T$, with probability exceeding $1 - O(T^{-10})$ we have*

$$N_{t_2}(a_+, a_-) - N_{t_1}(a_+, a_-) \leq C_5 \sum_{t=t_1+1}^{t_2} \frac{\pi_{\mathsf{ref}}(a_-)}{\pi_{\mathsf{ref}}(a_+)} \left[ \mathsf{KL}\big(\sigma(r^\star(a_+) - r^\star(a_-)) \| \sigma(r^{(t)}(a_+) - r^{(t)}(a_-))\big)^{\frac{1}{\beta}} \right.$$
$$\left. + \sigma(r^\star(a_-) - r^\star(a_+))^{\frac{1}{\beta}} \right] + C_5 \sqrt{T \log T}.$$

Equipped with Lemma 3, we can bound each $\zeta(a_+, a_-)$ using both density ratios regarding human feedback $\pi_{\mathsf{HF}}(a_+)/\pi_{\mathsf{HF}}(a_-)$, and regarding the reference policy $\pi_{\mathsf{ref}}(a_-)/\pi_{\mathsf{ref}}(a_+)$. The proof is deferred to Appendix C.4.

**Lemma 4.** *There exists universal constant $C_6 > 0$ such that, for any action pair $(a_+, a_-)$, with probability exceeding $1 - O(T^{-9})$ we have*

$$\zeta(a_+, a_-) \leq C_6 (r_{\max} + \log T) \min \left\{ \frac{\pi_{\mathsf{HF}}(a_+)}{\pi_{\mathsf{HF}}(a_-)} \alpha_T r_{\max}, \left( T \frac{\pi_{\mathsf{ref}}(a_-)}{\pi_{\mathsf{ref}}(a_+)} \right)^{\frac{\beta}{\beta+1}} \alpha_T^{\frac{1}{\beta+1}} r_{\max}^{\frac{1}{\beta+1}} \right\}$$
$$+ C_6 \left( \frac{A N_T(a_+, a_-) \log T}{\alpha_T} + \sqrt{T \log T} \right) r_{\max}.$$

This immediately implies that

$$\zeta \leq 2\tau A^2 r_{\max} + C_6 \left( \frac{AT \log T}{\alpha_T} + A^2 \sqrt{T \log T} \right) r_{\max} \tag{5.9}$$

$$+ C_6 (r_{\max} + \log T) \sum_{r^\star(a_+) \geq r^\star(a_-)} \min \left\{ \frac{\pi_{\mathsf{HF}}(a_+)}{\pi_{\mathsf{HF}}(a_-)} \alpha_T r_{\max}, \left( T \frac{\pi_{\mathsf{ref}}(a_-)}{\pi_{\mathsf{ref}}(a_+)} \right)^{\frac{\beta}{\beta+1}} \alpha_T^{\frac{1}{\beta+1}} r_{\max}^{\frac{1}{\beta+1}} \right\}.$$

Putting the regret decomposition (5.3) and the bounds (5.4), (5.5) and (5.9) collectively yields the desired regret bound (4.1).

# 6 DISCUSSION

In this paper, we investigated the problem of efficient exploration in online RLHF. By a careful analysis of the existing optimism-based exploration strategies, we identified a conceptual drawback in their sampling protocol, and we proved lower bounds to show that they can lead to inefficient exploration. We then proposed our algorithm that explicitly targets uncertainty in reward differences most relevant for policy improvement. Under a multi-armed bandit setup of RLHF, we establish regret bounds of order $T^{(\beta+1)/(\beta+2)}$, which scales polynomially in all model parameters.

Our work opens several avenues for future investigation. An immediate question is whether the rate $T^{(\beta+1)/(\beta+2)}$ is minimax optimal, or if faster rates can be achieved. Another important direction is to refine the dependence on parameters such as $A$ and $r_{\max}$, which may be improved with sharper analysis or alternative exploration schemes. Finally, our theoretical results are restricted to the bandit setting; extending the analysis to richer environments that incorporate a prompt space would be an exciting step toward bridging theory and practice in online RLHF.

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

LLM USAGE

In preparing this paper, large language models (LLMs) were used as an assistive tool for minor language polishing. All technical contributions, results, and conclusions are solely the work of the authors.

## A  PROOF OF PROPOSITION 1

For each $t \geq 1$, define the event

$$\mathcal{E}_t := \{\text{no } a_0 \text{ is sampled in the first } t \text{ samples}\}.$$

We will show that for any $t \geq 1$,

$$\mathbb{P}(\mathcal{E}_t) \geq \frac{4}{9}\big(1 - \exp(-r_{\max}/\beta)\big)^{2(t-1)}. \tag{A.1}$$

Conditional on $\mathcal{E}_t$, it can be seen that $\ell(r, \mathcal{D}^{(t)})$ only depends on $r(a_1) - r(a_2)$. Now we study when we fix $r(a_1) - r(a_2) \equiv \delta$ such that $\ell(r, \mathcal{D}^{(t)})$ is fixed, when is $J(\pi, r; \pi_{\text{cal}})$ maximized over both $\pi$ and $r$. By symmetry, we can assume without loss of generality that $\delta \geq 0$. We can compute

$$\begin{aligned}
J(\pi, r; \pi_{\text{cal}}) &= \mathbb{E}_{a \sim \pi}[r(a)] - \mathbb{E}_{a \sim \pi_{\text{cal}}}[r(a)] - \beta \mathsf{KL}(\pi \parallel \pi_{\text{ref}}) \\
&= [\pi(a_1) - p][r(a_1) - r(a_0)] + [\pi(a_2) - p][r(a_2) - r(a_0)] - \beta \mathsf{KL}(\pi \parallel \pi_{\text{ref}}) \\
&= [\pi(a_1) + \pi(a_2) - 2p][r(a_1) - r(a_0)] - \delta[\pi(a_2) - p] - \beta \mathsf{KL}(\pi \parallel \pi_{\text{ref}}).
\end{aligned}$$

For fixed $\pi$, we check which reward function $r$ maximizes $J(\pi, r; \pi_{\text{cal}})$.

- When $\pi(a_1) + \pi(a_2) > 2p$, we know that

$$\max_r J(\pi, r; \pi_{\text{cal}}) = r_{\max}[\pi(a_1) + \pi(a_2) - 2p] - \delta[\pi(a_2) - p] - \beta \mathsf{KL}(\pi \parallel \pi_{\text{ref}}), \tag{A.2}$$

which is maximized at $r(a_1) = r_{\max}$, $r(a_2) = r_{\max} - \delta$ and $r(a_0) = 0$.

- When $\pi(a_1) + \pi(a_2) < 2p$, we know that

$$\max_r J(\pi, r; \pi_{\text{cal}}) = (r_{\max} - \delta)[2p - \pi(a_1) - \pi(a_2)] - \delta[\pi(a_2) - p] - \beta \mathsf{KL}(\pi \parallel \pi_{\text{ref}}), \tag{A.3}$$

which is maximized at $r(a_1) = \delta$, $r(a_2) = 0$ and $r(a_0) = r_{\max}$.

In addition, for any policy $\pi$ such that $\pi(a_1) + \pi(a_2) < 2p$, by considering another policy $\pi'$ defined as $\pi'(a_1) = 2p - \pi(a_2)$ and $\pi'(a_2) = 2p - \pi(a_1)$, we have

$$\begin{aligned}
\max_r &J(\pi', r; \pi_{\text{cal}}) - \max_r J(\pi, r; \pi_{\text{cal}}) \\
&= r_{\max}[\pi'(a_1) + \pi'(a_2) - 2p] - \delta[\pi'(a_2) - p] - \beta \mathsf{KL}(\pi' \parallel \pi_{\text{ref}}) \\
&\quad - (r_{\max} - \delta)[2p - \pi(a_1) - \pi(a_2)] + \delta[\pi(a_2) - p] + \beta \mathsf{KL}(\pi \parallel \pi_{\text{ref}}) \\
&= \beta[\mathsf{KL}(\pi \parallel \pi_{\text{ref}}) - \mathsf{KL}(\pi' \parallel \pi_{\text{ref}})].
\end{aligned}$$

Here the first relation follows from (A.2), (A.3) and the fact that $\pi'(a_1) + \pi'(a_2) > 2p$. Let $x = \pi(a_1)$ and $y = \pi(a_2)$. Let

$$\begin{aligned}
f(x, y) &:= \mathsf{KL}(\pi \parallel \pi_{\text{ref}}) - \mathsf{KL}(\pi' \parallel \pi_{\text{ref}}) \\
&= x \log x + y \log y + (1 - x - y) \log(1 - x - y) - (2p - x) \log(2p - x) \\
&\quad - (2p - y) \log(2p - y) - (1 - 4p + x + y) \log(1 + x + y - 4p).
\end{aligned}$$

By elementary analysis, it is straightforward to check that $f(x, y) > 0$ for any $x, y > 0$ satisfying $x + y < 2p$. Therefore we have

$$\max_r J(\pi', r; \pi_{\text{cal}}) > \max_r J(\pi, r; \pi_{\text{cal}}).$$

Therefore in order to maximize $\ell(r, \mathcal{D}^{(t)}) + \alpha J^{\star}(r; \pi_{\text{cal}})$, the following statement always holds regardless of the value of $\delta$:

$$r^{(t+1)}(a_0) = 0, \quad \max\{r^{(t+1)}(a_1), r^{(t+1)}(a_2)\} = r_{\max}.$$

This immediately implies that

$$\pi^{(t+1)}(a_0) = \frac{\exp(r^{(t+1)}(a_0)/\beta)}{\exp(r^{(t+1)}(a_0)/\beta) + \exp(r^{(t+1)}(a_1)/\beta) + \exp(r^{(t+1)}(a_2)/\beta)} \leq \frac{1}{2 + \exp(r_{\max}/\beta)}.$$

Therefore conditional on $\mathcal{E}_t$, we know that

$$\mathbb{P}(\mathcal{E}_{t+1}|\mathcal{E}_t) \geq \left(1 - \pi^{(t+1)}(a_0)\right)^2 \geq \left(\frac{1}{1 + \exp(-r_{\max}/\beta)}\right)^2 \geq \left(1 - \exp(-r_{\max}/\beta)\right)^2.$$

This relation, together with

$$\mathbb{P}(\mathcal{E}_0) = \left(\pi^{(1)}(a_1) + \pi^{(1)}(a_2)\right)^2 = \frac{4}{9},$$

establishes the statement (A.1). This immediately implies that, for any $t \leq \exp(r_{\max}/\beta)/2$,

$$\mathbb{P}(\mathcal{E}_t) \geq \frac{4}{9}\left(1 - \exp(-r_{\max}/\beta)\right)^{2(t-1)} \geq \frac{4}{9}\left(1 - \exp(-r_{\max}/\beta)\right)^{\exp(r_{\max}/\beta)} \geq \frac{4}{9e} \geq 0.16.$$

Finally, when $\mathcal{E}_t$ holds, we have

$$J(\pi^\star; r^\star, \pi_{\mathsf{cal}}) - J(\pi^{(t)}; r^\star, \pi_{\mathsf{cal}}) = \pi^\star(a_0) - \pi^{(t)}(a_0) - \beta\mathsf{KL}(\pi^\star\|\pi_{\mathsf{ref}}) + \beta\mathsf{KL}(\pi^{(t)}\|\pi_{\mathsf{ref}}).$$

We have

$$\pi^\star(a_0) = \frac{\exp(1/\beta)}{\exp(1/\beta) + 2}, \quad \pi^\star(a_1) = \pi^\star(a_2) = \frac{1}{\exp(1/\beta) + 2}.$$

Therefore we have

$$\begin{aligned}
\mathsf{KL}(\pi^\star \| \pi_{\mathsf{ref}}) &= \log 3 + \pi^\star(a_0)\log\pi^\star(a_0) + \pi^\star(a_1)\log\pi^\star(a_1) + \pi^\star(a_2)\log\pi^\star(a_2) \\
&= \log 3 + \frac{\exp(1/\beta)}{\exp(1/\beta) + 2}\log\frac{\exp(1/\beta)}{\exp(1/\beta) + 2} + \frac{2}{\exp(1/\beta) + 2}\log\frac{1}{\exp(1/\beta) + 2} \\
&= \log 3 + \beta^{-1}\frac{\exp(1/\beta)}{\exp(1/\beta) + 2} - \log[\exp(1/\beta) + 2].
\end{aligned}$$

In addition, when $\mathcal{E}^{t-1}$ happens, we know that

$$\begin{aligned}
\mathsf{KL}(\pi^{(t)} \| \pi_{\mathsf{ref}}) &= \log 3 + \pi^{(t)}(a_0)\log\pi^{(t)}(a_0) + \pi^{(t)}(a_1)\log\pi^{(t)}(a_1) + \pi^{(t)}(a_2)\log\pi^{(t)}(a_2) \\
&\overset{(i)}{\geq} \log 3 + \pi^{(t)}(a_0)\log\pi^{(t)}(a_0) + \left[\pi^{(t)}(a_1) + \pi^{(t)}(a_2)\right]\log\frac{\pi^{(t)}(a_1) + \pi^{(t)}(a_2)}{2} \\
&= \log 3 + \pi^{(t)}(a_0)\log\pi^{(t)}(a_0) + \left[1 - \pi^{(t)}(a_0)\right]\log\frac{1 - \pi^{(t)}(a_0)}{2} \\
&\overset{(ii)}{\geq} \log 3 - \log 2 - 0.16.
\end{aligned}$$

Here step (i) uses Jensen's inequality for convex function $f(x) = x\log x$; step (ii) holds since the function $g(x) = x\log x + (1-x)\log(1-x)/2$ is monotonically decreasing for $0 < x < 1/3$, and we have

$$\pi^{(t)}(a_0) \leq \frac{1}{2 + \exp(r_{\max}/\beta)} \leq \frac{1}{2 + \exp(3)} \leq 0.046$$

provided that $r_{\max}/\beta \geq 3$. We have

$$\begin{aligned}
J(\pi^\star; r^\star, \pi_{\mathsf{cal}}) - J(\pi^{(t)}; r^\star, \pi_{\mathsf{cal}}) &= \pi^\star(a_0) - \pi^{(t)}(a_0) - \beta\mathsf{KL}(\pi^\star \| \pi_{\mathsf{ref}}) + \beta\mathsf{KL}(\pi^{(t)} \| \pi_{\mathsf{ref}}) \\
&\geq \beta\log\left(\exp(1/\beta) + 2\right) - (\log 2 + 0.16)\beta - 0.046 \\
&\geq 1/2,
\end{aligned}$$

where the last relation holds for any $\beta > 0$.

## B  PROOF OF PROPOSITION 2

Let $T = \min\{\kappa, \exp(r_{\max})/2\}$, and define the events

$$\mathcal{A} := \left\{a_2^t = a_0 \text{ for all } 1 \leq t \leq T\right\}$$

and

$$\mathcal{E} := \{a_1^t \succ a_2^t \text{ or } a_1^t = a_2^t \text{ for all } 1 \leq t \leq T\}.$$

We can check that when $\kappa \geq 5$,

$$\mathbb{P}(\mathcal{A}) = [\pi_{\mathsf{ref}}(a_0)]^T \leq (1 - 2\kappa^{-1})^\kappa \geq \frac{1}{16}.$$

Conditional on $\mathcal{A}$, we know that when $r_{\max} \geq 1$,

$$\mathbb{P}(\mathcal{E} \mid \mathcal{A}) \geq \left(\frac{\exp(r_{\max} - 1)}{1 + \exp(r_{\max} - 1)}\right)^T \geq \left(\frac{\exp(r_{\max} - 1)}{1 + \exp(r_{\max} - 1)}\right)^{\exp(r_{\max})/2} \geq \frac{1}{4}.$$

Conditional on $\mathcal{A}$ and $\mathcal{E}_t$, for any $1 \leq t \leq T-1$, all the preference data in $\mathcal{D}^{(t)}$ are of form $a_1^t \succ a_2^t$. In this case, it is straightforward to check that the reward function that maximizes $\ell(r, \mathcal{D}^{(t)}) + \alpha J^\star(r; \pi_{\mathsf{ref}})$ is

$$r^{(t+1)}(a_0) = 0, \quad r^{(t+1)}(a_1) = r^{(t+1)}(a_2) = r_{\max}.$$

This immediately implies that

$$\pi^{(t+1)}(a_0) = \frac{\kappa - 2}{\kappa - 2 + 2\exp(r_{\max}/\beta)}, \quad \pi^{(t+1)}(a_1) = \pi^{(t+1)}(a_2) = \frac{\exp(r_{\max}/\beta)}{\kappa - 2 + 2\exp(r_{\max}/\beta)}.$$

On the other hand, we know that

$$\pi^\star(a_0) = \frac{\kappa - 2}{\kappa - 2 + \exp(r_{\max}/\beta) + \exp((r_{\max} - 2)/\beta)},$$

$$\pi^\star(a_1) = \frac{\exp(r_{\max}/\beta)}{\kappa - 2 + \exp(r_{\max}/\beta) + \exp((r_{\max} - 2)/\beta)},$$

$$\pi^\star(a_2) = \frac{\exp((r_{\max} - 2)/\beta)}{\kappa - 2 + \exp(r_{\max}/\beta) + \exp((r_{\max} - 2)/\beta)}.$$

For any $2 \leq t \leq T$, we first lower bound

$$J(\pi^\star; r^\star, \pi_{\mathsf{ref}}) - J(\pi^{(t)}; r^\star, \pi_{\mathsf{ref}}) \geq J(\pi_{\theta^\star}; r^\star, \pi_{\mathsf{ref}}) - J(\pi_1; r^\star, \pi_{\mathsf{ref}}) \tag{B.1}$$

for any $\theta^\star \in [0, 1]$, where we define $\pi_\theta := \theta\pi^{(t)} + (1 - \theta)\pi^\star$, and the above relation follows from the optimality of $\pi^\star$. Recall the definition

$$J(\pi; r^\star, \pi_{\mathsf{ref}}) = \pi(a_1)r_{\max} + \pi(a_2)(r_{\max} - 2) - \beta \sum_{i=0}^{2} \pi(a_i) \log \frac{\pi(a_i)}{\pi_{\mathsf{ref}}(a_i)},$$

we can compute

$$\nabla_\pi J(\pi; r^\star, \pi_{\mathsf{ref}}) = \begin{bmatrix} r^\star(a_0) - \beta \log[\pi(a_0)/\pi_{\mathsf{ref}}(a_0)] - \beta \\ r^\star(a_1) - \beta \log[\pi(a_1)/\pi_{\mathsf{ref}}(a_1)] - \beta \\ r^\star(a_2) - \beta \log[\pi(a_2)/\pi_{\mathsf{ref}}(a_2)] - \beta \end{bmatrix}$$

and

$$\nabla_\pi^2 J(\pi; r^\star, \pi_{\mathsf{ref}}) = -\beta \mathsf{diag}\left\{\pi(a_0), \pi(a_1), \pi(a_2)\right\}^{-1}.$$

It is straightforward to check that

$$\nabla_\pi J(\pi^{(t)}; r^\star, \pi_{\mathsf{ref}}) = \begin{bmatrix} 0 \\ 0 \\ -1 \end{bmatrix} + \mathsf{const} \cdot \begin{bmatrix} 1 \\ 1 \\ 1 \end{bmatrix}. \tag{B.2}$$

Since $\pi^{(t)}(a_0) < \pi^\star(a_0)$, $\pi^{(t)}(a_1) < \pi^\star(a_1)$ and $\pi^{(t)}(a_2) > \pi^\star(a_2)$, we know that for any $\theta \in [0, \theta^\star]$

$$\nabla_\pi^2 J(\pi_\theta; r^\star, \pi_{\mathsf{ref}}) \succeq -\beta \mathsf{diag}\big\{\pi^{(t)}(a_0), \pi^{(t)}(a_1), \theta^\star \pi^{(t)}(a_2) + (1-\theta^\star)\pi^\star(a_2)\big\}^{-1}. \tag{B.3}$$

Therefore we have

$$
\begin{aligned}
J(\pi_{\theta^\star}; r^\star, \pi_{\mathsf{ref}}) - J(\pi_1; r^\star, \pi_{\mathsf{ref}}) &\overset{(i)}{\geq} \theta^\star \nabla_\pi J(\pi^{(t)}; r^\star, \pi_{\mathsf{ref}})^\top (\pi^\star - \pi^{(t)}) \\
&\quad - \frac{\beta\theta^{\star2}}{2}(\pi^\star - \pi^{(t)})^\top \mathsf{diag}\big\{\pi^{(t)}(a_0), \pi^{(t)}(a_1), \theta^\star \pi^{(t)}(a_2) + (1-\theta^\star)\pi^\star(a_2)\big\}^{-1}(\pi^\star - \pi^{(t)}) \\
&\overset{(ii)}{\geq} \theta^\star[\pi^{(t)}(a_2) - \pi^\star(a_2)] - \frac{\beta\theta^{\star2}}{2}\Big[\pi^\star(a_0) + \pi^\star(a_1) + \frac{9}{16}\pi^{(t)}(a_2)/\theta^\star\Big] \\
&= \theta^\star[\pi^{(t)}(a_2) - \pi^\star(a_2)] - \frac{9}{32}\beta\theta^\star\pi^{(t)}(a_2) - \frac{\beta\theta^{\star2}}{2}[1 - \pi^\star(a_2)] \\
&= \Big(1 - \frac{9}{32}\beta\Big)\theta^\star\pi^{(t)}(a_2) - \Big(1 - \frac{\beta\theta^\star}{2}\Big)\theta^\star\pi^\star(a_2) - \frac{\beta\theta^{\star2}}{2} \\
&\overset{(iii)}{\geq} \Big(\frac{3}{4} - \frac{9}{32}\beta + \frac{\beta\theta^\star}{8}\Big)\theta^\star\pi^{(t)}(a_2) - \frac{\beta\theta^{\star2}}{2} \overset{(iv)}{\geq} \frac{15}{32}\theta^\star\pi^{(t)}(a_2) - \frac{\theta^{\star2}}{2}. \tag{B.4}
\end{aligned}
$$

Here step (i) follows from the Taylor expansion and (B.3); step (ii) utilizes (B.2) and as well as the following relations

$$\pi^{(t)}(a_0) \leq \pi^\star(a_0) \leq 2\pi^{(t)}(a_0), \quad \pi^{(t)}(a_1) \leq \pi^\star(a_1) \leq 2\pi^{(t)}(a_1)$$

and when $\beta \leq 1$,

$$\pi^\star(a_2) \leq \frac{2}{\exp(2/\beta) + 1}\pi^{(t)}(a_2) \leq \frac{1}{4}\pi^{(t)}(a_2); \tag{B.5}$$

steps (iii) and (iv) follows from (B.5) and $\beta \leq 1$. When $\kappa \leq \exp(r_{\max}\beta)$, we have

$$\pi^{(t)}(a_2) = \frac{\exp(r_{\max}/\beta)}{\kappa - 2 + 2\exp(r_{\max}/\beta)} \geq \frac{\exp(r_{\max}/\beta)}{3\exp(r_{\max}/\beta) - 2} \geq \frac{1}{3}. \tag{B.6}$$

By taking (B.1), (B.4) and (B.6) collectively, we have

$$(\pi^\star; r^\star, \pi_{\mathsf{ref}}) - J(\pi^{(t)}; r^\star, \pi_{\mathsf{ref}}) \geq \frac{5}{32}\theta^\star - \frac{\theta^{\star2}}{2} \geq \frac{25}{2048} > 0.01$$

where we take $\theta^\star = 5/32$.

## C PROOF OF AUXILIARY LEMMAS

### C.1 PROOF OF LEMMA 1

We first express

$$X_i := \log \frac{\sigma(r^\star(a_+^i) - r^\star(a_-^i))}{\sigma(r(a_+^i) - r(a_-^i))} = \mathbb{1}\{a_1^i \succ a_2^i\}\log \frac{\sigma(r^\star(a_1^i) - r^\star(a_2^i))}{\sigma(r(a_1^i) - r(a_2^i))} + \mathbb{1}\{a_1^i \prec a_2^i\}\log \frac{\sigma(r^\star(a_2^i) - r^\star(a_1^i))}{\sigma(r(a_2^i) - r(a_1^i))}.$$

It is straightforward to check that

$$
\begin{aligned}
\mathbb{E}\big[X_i \big| a_1^i, a_2^i\big] &= \mathbb{P}\big(a_1^i \succ a_2^i \big| a_1^i, a_2^i\big)\log \frac{\sigma(r^\star(a_1^i) - r^\star(a_2^i))}{\sigma(r(a_1^i) - r(a_2^i))} + \mathbb{P}\big(a_1^i \prec a_2^i \big| a_1^i, a_2^i\big)\log \frac{\sigma(r^\star(a_2^i) - r^\star(a_1^i))}{\sigma(r(a_2^i) - r(a_1^i))} \\
&= \sigma(r^\star(a_1^i) - r^\star(a_2^i))\log \frac{\sigma(r^\star(a_1^i) - r^\star(a_2^i))}{\sigma(r(a_1^i) - r(a_2^i))} + \sigma(r^\star(a_2^i) - r^\star(a_1^i))\log \frac{\sigma(r^\star(a_2^i) - r^\star(a_1^i))}{\sigma(r(a_2^i) - r(a_1^i))} \\
&= \mathsf{KL}\big(\sigma(r^\star(a_1^i) - r^\star(a_2^i)) \,\|\, \sigma(r(a_1^i) - r(a_2^i))\big).
\end{aligned}
$$

and

$$|X_i| \leq \big|\log\big(1 + \exp(-r(a_+^i) + r(a_-^i))\big)\big| \leq 2r_{\max}.$$

In addition, we can compute the variance

$$\mathsf{Var}\left(X_i\big|a_1^i, a_2^i\right) = \sigma(r^\star(a_1^i) - r^\star(a_2^i))\sigma(r^\star(a_2^i) - r^\star(a_1^i))\left[\log\frac{\sigma(r^\star(a_1^i) - r^\star(a_2^i))}{\sigma(r(a_1^i) - r(a_2^i))} - \log\frac{\sigma(r^\star(a_2^i) - r^\star(a_1^i))}{\sigma(r(a_2^i) - r(a_1^i))}\right]^2$$

$$= \sigma(r^\star(a_1^i) - r^\star(a_2^i))\sigma(r^\star(a_2^i) - r^\star(a_1^i))\left[\log\frac{\sigma(r^\star(a_1^i) - r^\star(a_2^i))}{\sigma(r^\star(a_2^i) - r^\star(a_1^i))} - \log\frac{\sigma(r(a_1^i) - r(a_2^i))}{\sigma(r(a_2^i) - r(a_1^i))}\right]^2$$

$$= \sigma(r^\star(a_1^i) - r^\star(a_2^i))\sigma(r^\star(a_2^i) - r^\star(a_1^i))\big[r(a_1^i) - r(a_2^i) - r^\star(a_1^i) + r^\star(a_2^i)\big]^2.$$

In view of Lemma 5, we have

$$\mathsf{KL}\big(\sigma(r^\star(a_1^i) - r^\star(a_2^i)) \parallel \sigma(r(a_1^i) - r(a_2^i))\big)$$

$$\geq \frac{1}{4}\sigma(r^\star(a_1^i) - r^\star(a_2^i))\sigma(r^\star(a_2^i) - r^\star(a_1^i))$$

$$\cdot \min\left\{|r(a_1^i) - r(a_2^i) - r^\star(a_1^i) + r^\star(a_2^i)|, \big[r(a_1^i) - r(a_2^i) - r^\star(a_1^i) + r^\star(a_2^i)\big]^2\right\}$$

$$\geq \frac{1}{16r_{\max}}\sigma(r^\star(a_1^i) - r^\star(a_2^i))\sigma(r^\star(a_2^i) - r^\star(a_1^i))\big[r(a_1^i) - r(a_2^i) - r^\star(a_1^i) + r^\star(a_2^i)\big]^2,$$

$$\text{(C.1)}$$

where the last step follows from $|r(a_1^i) - r(a_2^i) - r^\star(a_1^i) + r^\star(a_2^i)| \leq 4r_{\max}$. Therefore we have

$$\mathsf{Var}\left(X_i\big|a_1^i, a_2^i\right) \leq 16r_{\max}\mathsf{KL}\big(\sigma(r^\star(a_+^i) - r^\star(a_-^i)) \parallel \sigma(r(a_+^i) - r(a_-^i))\big).$$

In addition, we have the following deterministic bound

$$\sum_{i=1}^{t}\mathsf{Var}\left(X_i\big|a_1^i, a_2^i\right) \leq 16tr_{\max}^2.$$

By the Freedman's inequality (cf. Lemma 6), for any fixed $r$, with probability exceeding $1 - \delta$,

$$|\Delta_t(r)| \leq \left|\sum_{i=1}^{t}\left(X_i - \mathbb{E}\left[X_i\big|a_1^i, a_2^i\right]\right)\right|$$

$$\leq C_2\sqrt{\sum_{i=1}^{t}r_{\max}\mathsf{KL}\big(\sigma(r^\star(a_1^i) - r^\star(a_2^i)) \parallel \sigma(r(a_1^i) - r(a_2^i))\big)\log\frac{\log t}{\delta}} + C_2 r_{\max}\log\frac{\log t}{\delta}$$

for some sufficiently large constant $C_2 > 0$.

## C.2 PROOF OF LEMMA 2

For any fixed $r : \mathcal{A} \to [\pm r_{\max}]$, with probability exceeding $1 - \delta$ we have

$$|\Delta_t(r)| \overset{(i)}{\leq} C_2\sqrt{\sum_{i=1}^{t}r_{\max}\mathsf{KL}\big(\sigma(r^\star(a_1^i) - r^\star(a_2^i)) \parallel \sigma(r(a_1^i) - r(a_2^i))\big)\log\frac{\log T}{\delta}} + C_2 r_{\max}\log\frac{\log T}{\delta}$$

$$\overset{(ii)}{\leq} \frac{1}{2}\sum_{i}\mathsf{KL}\big(\sigma(r^\star(a_1^i) - r^\star(a_2^i)) \parallel \sigma(r(a_1^i) - r(a_2^i))\big) + 2C_2^2 r_{\max}\log\frac{\log T}{\delta}.$$

Here step (i) follows from Lemma 1, and step (ii) utilizes the AM-GM inequality. This immediately implies that

$$\ell(r, \mathcal{D}^{(t)}) - \ell(r^\star, \mathcal{D}^{(t)}) = \sum_{i=1}^{t}\log\frac{\sigma(r(a_+^i) - r(a_-^i))}{\sigma(r^\star(a_+^i) - r^\star(a_-^i))}$$

$$= -\sum_{i=1}^{t}\mathsf{KL}\big(\sigma(r^\star(a_1^i) - r^\star(a_2^i)) \parallel \sigma(r(a_1^i) - r(a_2^i))\big) - \Delta_t(r)$$

$$\leq -\frac{1}{2}\sum_{i=1}^{t}\mathsf{KL}\big(\sigma(r^\star(a_1^i) - r^\star(a_2^i)) \,\|\, \sigma(r(a_1^i) - r(a_2^i))\big) + 2C_2^2 r_{\max}\log\frac{\log T}{\delta}. \tag{C.2}$$

Then we explore the Lipschitzness continuity of the above functionals of $r$. For any two fixed reward functions $r, r' : \mathcal{A} \to [\pm r_{\max}]$, we have

$$\Big|\ell(r, \mathcal{D}^{(t)}) - \ell(r', \mathcal{D}^{(t)})\Big| = \sum_{i=1}^{t}\Big|\log[\sigma(r(a_+^i) - r(a_-^i))] - \log[\sigma(r'(a_+^i) - r'(a_-^i))]\Big|$$

$$\leq \sum_{i=1}^{t}|r(a_+^i) - r(a_-^i) - r'(a_+^i) + r'(a_-^i)| \leq 2T\|r - r'\|_\infty, \quad \text{(C.3)}$$

where the penultimate step follows from $\mathrm{d}\log(\sigma(x))/\mathrm{d}x = \sigma(-x) \leq 1$. Similarly, for any $x, y, \delta \in \mathbb{R}$, we have

$$\big|\mathsf{KL}\big(\sigma(x) \,\|\, \sigma(y)\big) - \mathsf{KL}\big(\sigma(x) \,\|\, \sigma(y + \delta)\big)\big| = \Big|\sigma(x)\log\frac{\sigma(y+\delta)}{\sigma(y)} + (1 - \sigma(x))\log\frac{1 - \sigma(y + \delta)}{1 - \sigma(y)}\Big|$$

$$\leq \sigma(x)|\delta| + (1 - \sigma(x))|\delta| = |\delta|.$$

This implies that

$$\Big|\sum_{i=1}^{t}\mathsf{KL}\big(\sigma(r^\star(a_1^i) - r^\star(a_2^i)) \,\|\, \sigma(r(a_1^i) - r(a_2^i))\big)$$

$$- \sum_{i=1}^{t}\mathsf{KL}\big(\sigma(r^\star(a_1^i) - r^\star(a_2^i)) \,\|\, \sigma(r'(a_1^i) - r'(a_2^i))\big)\Big| \leq 2\|r - r'\|_\infty. \tag{C.4}$$

Let $\mathcal{N}_\varepsilon$ be an $\varepsilon$-net of $[-r_{\max}, r_{\max}]^A$ (or equivalently, the function space of $r : \mathcal{A} \to [\pm r_{\max}]$) under the $\ell_\infty$ norm such that $|\mathcal{N}_\varepsilon| \leq (2r_{\max}/\varepsilon)^A$. By standard union bound argument and (C.2), with probability exceeding $1 - \delta$,

$$\ell(r, \mathcal{D}^{(t)}) - \ell(r^\star, \mathcal{D}^{(t)}) \leq -\frac{1}{2}\sum_{i=1}^{t}\mathsf{KL}\big(\sigma(r^\star(a_1^i) - r^\star(a_2^i)) \,\|\, \sigma(r(a_1^i) - r(a_2^i))\big) + 2C_2^2 r_{\max}\log\frac{|\mathcal{N}_\varepsilon|\log T}{\delta} \tag{C.5}$$

holds for any $r \in \mathcal{N}_\varepsilon$. This implies that for any $r : \mathcal{A} \to [\pm r_{\max}]$, there exists $r_0 \in \mathcal{N}_\varepsilon$ such that $\|r - r'\| \leq \varepsilon$, hence

$$\ell(r, \mathcal{D}^{(t)}) - \ell(r^\star, \mathcal{D}^{(t)}) \overset{\text{(i)}}{\leq} \ell(r_0, \mathcal{D}^{(t)}) - \ell(r^\star, \mathcal{D}^{(t)}) + 2T\varepsilon$$

$$\overset{\text{(ii)}}{\leq} -\frac{1}{2}\sum_{i=1}^{t}\mathsf{KL}\big(\sigma(r^\star(a_1^i) - r^\star(a_2^i)) \,\|\, \sigma(r_0(a_1^i) - r_0(a_2^i))\big) + 2C_2^2 r_{\max}\log\frac{|\mathcal{N}_\varepsilon|\log T}{\delta} + 2T\varepsilon$$

$$\overset{\text{(iii)}}{\leq} -\frac{1}{2}\sum_{i=1}^{t}\mathsf{KL}\big(\sigma(r^\star(a_1^i) - r^\star(a_2^i)) \,\|\, \sigma(r(a_1^i) - r(a_2^i))\big) + 2C_2^2 r_{\max}\log\frac{|\mathcal{N}_\varepsilon|\log T}{\delta} + 4T\varepsilon$$

$$\overset{\text{(iv)}}{\leq} -\frac{1}{2}\sum_{i=1}^{t}\mathsf{KL}\big(\sigma(r^\star(a_1^i) - r^\star(a_2^i)) \,\|\, \sigma(r(a_1^i) - r(a_2^i))\big) + C_3 A r_{\max}\log T.$$

Here step (i) utilizes (C.3); step (ii) follows from $r_0 \in \mathcal{N}_\varepsilon$ and the uniform concentration bound (C.5); step (iii) uses (C.4); step (iv) holds as long as $C_3 \gg 2C_2^2$, where we let $\varepsilon = A r_{\max}/T$ and $\delta = T^{-10}$. This completes the proof.

### C.3 PROOF OF LEMMA 3

When $N_t(a_+, a_-) \geq 100 C_4 \alpha_t r_{\max}$, we have

$$\mathsf{KL}\big(\sigma(r^\star(a_+) - r^\star(a_-)) \,\|\, \sigma(r^{(t)}(a_+) - r^{(t)}(a_-))\big) \leq \frac{1}{100}. \tag{C.6}$$

Now we assert that $r^{(t)}(a_-) - r^{(t)}(a_+) < 0.5$ for any $t \geq t_0$. This is because, if $r^{(t)}(a_-) - r^{(t)}(a_+) \geq 0.5$, we have

$$\mathsf{KL}\big(\sigma(r^\star(a_+) - r^\star(a_-)) \,\|\, \sigma(r^{(t)}(a_+) - r^{(t)}(a_-))\big) = \mathsf{KL}\big(\sigma(r^\star(a_-) - r^\star(a_+)) \,\|\, \sigma(r^{(t)}(a_-) - r^{(t)}(a_+))\big)$$

$$\geq \mathsf{KL}\big(\sigma(0) \,\|\, \sigma(0.5)\big) > \frac{1}{100}.$$

Here we use the fact that $r^\star(a_-) - r^\star(a_+) \leq 0$. This contradicts with (C.6). Hence we have

$$r^{(t)}(a_-) - r^{(t)}(a_+) < 0.5. \tag{C.7}$$

Let $p := \sigma(r^\star(a_-) - r^\star(a_+))$ and $q := \sigma(r^{(t)}(a_-) - r^{(t)}(a_+))$. We have

$$\exp(r^{(t)}(a_-) - r^{(t)}(a_+)) \overset{(i)}{\leq} 3\sigma(r^{(t)}(a_-) - r^{(t)}(a_+)) = 3q \overset{(ii)}{\leq} 6p + \mathsf{KL}(p \,\|\, q) \tag{C.8}$$

$$= 6\sigma(r^\star(a_-) - r^\star(a_+)) + 24\mathsf{KL}\big(\sigma(r^\star(a_+) - r^\star(a_-)) \,\|\, \sigma(r^{(t)}(a_+) - r^{(t)}(a_-))\big).$$

Here step (i) follows from (C.7), while step (ii) holds trivially when $q \leq 2p$, and when $q > 2p$ we have

$$\mathsf{KL}(p \,\|\, q) = p \log \frac{p}{q} + (1-p) \log \frac{1-p}{1-q} \geq \frac{(q-p)^2}{2q} \geq \frac{1}{8}q.$$

Finally, for any $t_0 \leq t_1 < t_2 \leq T$, we can upper bound

$$N_{t_2}(a_+, a_-) - N_{t_1}(a_+, a_-) \leq \sum_{i=t_1+1}^{t_2} X_i \quad \text{where} \quad X_i := \mathbb{1}\{a_- \text{ is sampled in the } i\text{-th iteration}\}.$$

It is straightforward to check that $X_i - \mathbb{E}[X_i|\mathcal{F}_{i-1}]$ is a martingale difference sequence, and by the Azuma-Hoeffding inequality, with probability exceeding $1 - O(T^{-100})$ we have

$$\sum_{i=t_1+1}^{t_2} \left( X_i - \mathbb{E}[X_i|\pi^{(i)}, \pi^{(i-1)}] \right) \leq \widetilde{C}\sqrt{T \log T}$$

for some universal constant $\widetilde{C} > 0$. In addition, we have

$$\mathbb{E}[X_i|\pi^{(i)}, \pi^{(i-1)}] \leq \frac{\pi^{(i)}(a_-)}{\pi^{(i)}(a_-) + \pi^{(i)}(a_+)} + \frac{\pi^{(i-1)}(a_-)}{\pi^{(i-1)}(a_-) + \pi^{(i-1)}(a_+)}.$$

For each $t \in [T]$, we have

$$\frac{\pi^{(t)}(a_-)}{\pi^{(t)}(a_-) + \pi^{(t)}(a_+)} \overset{(i)}{=} \frac{\pi_{\mathsf{ref}}(a_-)\exp(r^{(t)}(a_-)/\beta)}{\pi_{\mathsf{ref}}(a_-)\exp(r^{(t)}(a_-)/\beta) + \pi_{\mathsf{ref}}(a_+)\exp(r^{(t)}(a_+)/\beta)}$$

$$\leq \frac{\pi_{\mathsf{ref}}(a_-)}{\pi_{\mathsf{ref}}(a_+)} \exp\left( \big(r^{(t)}(a_-) - r^{(t)}(a_+)\big)/\beta \right)$$

$$\leq \frac{\pi_{\mathsf{ref}}(a_-)}{\pi_{\mathsf{ref}}(a_+)} \Big[ 6\sigma(r^\star(a_-) - r^\star(a_+)) + 24\mathsf{KL}\big(\sigma(r^\star(a_+) - r^\star(a_-)) \,\|\, \sigma(r^{(t)}(a_+) - r^{(t)}(a_-))\big) \Big]^{1/\beta}.$$

Here step (i) utilizes (2.4), while step (ii) follows from (C.8). Hence we have

$$N_{t_2}(a_+, a_-) - N_{t_1}(a_+, a_-) \leq 2 \sum_{t=t_1+1}^{t_2} \frac{\pi^{(t)}(a_-)}{\pi^{(t)}(a_-) + \pi^{(t)}(a_+)} + 2\widetilde{C}\sqrt{T \log T}$$

$$\leq C_5^{1/\beta} \sum_{t=t_1+1}^{t_2} \frac{\pi_{\mathsf{ref}}(a_-)}{\pi_{\mathsf{ref}}(a_+)} \Big[ \mathsf{KL}\big(\sigma(r^\star(a_+) - r^\star(a_-))\|\sigma(r^{(t)}(a_+) - r^{(t)}(a_-))\big)^{\frac{1}{\beta}}$$

$$+ \sigma(r^\star(a_-) - r^\star(a_+))^{\frac{1}{\beta}} \Big] + C_5\sqrt{T \log T}$$

for some sufficiently large constant $C_5 > 0$.

### C.4 PROOF OF LEMMA 4

Let $t_0$ be the first iteration such that

$$N_{t_0}(a_+, a_-) \geq \min\left\{ \frac{1}{2} N_T(a_+, a_-), 100 C_4 \alpha_T r_{\mathsf{max}} \right\}. \tag{C.9}$$

In what follows, we establish the desired result under two different cases: $N_T(a_+, a_-)$ being larger or smaller than $c_0 \exp(r^\star(a_+) - r^\star(a_-)) \alpha_T r_{\mathsf{max}}$ for some sufficiently large constant $c_0 > 0$.

**Case 1.** When $N_T(a_+, a_-) \leq c_0 \exp(r^\star(a_+) - r^\star(a_-)) \alpha_T r_{\mathsf{max}}$, it is straightforward to show that

$$\zeta(a_+, a_-) \leq N_T(a_+, a_-) r_{\mathsf{max}} \leq C_6 \exp(r^\star(a_+) - r^\star(a_-)) \alpha_T r_{\mathsf{max}}^2 = C_6 \frac{\pi_{\mathsf{HF}}(a_+)}{\pi_{\mathsf{HF}}(a_-)} \alpha_T r_{\mathsf{max}}^2. \tag{C.10}$$

In addition, we have

$$\sigma(r^\star(a_-) - r^\star(a_+)) \leq \exp(r^\star(a_-) - r^\star(a_+)) \leq \frac{c_0 \alpha_T r_{\mathsf{max}}}{N_T(a_+, a_-)}.$$

In addition, for any $t_0 \leq t \leq T$, we can use (5.8) to show that

$$\mathsf{KL}\big(\sigma(r^\star(a_+) - r^\star(a_-)) \| \sigma(r^{(t)}(a_+) - r^{(t)}(a_-))\big) \leq \frac{C_4 \alpha_t r_{\mathsf{max}}}{N_t(a_+, a_-)} \leq \frac{2 C_4 \alpha_T r_{\mathsf{max}}}{N_T(a_+, a_-)}. \tag{C.11}$$

By taking $t_1 = t_0 - 1$ and $t_2 = T$ in Lemma 3, we have

$$N_T(a_+, a_-) \overset{(i)}{\leq} 2[N_T(a_+, a_-) - N_{t_0-1}(a_+, a_-)]$$
$$\overset{(ii)}{\leq} 4T \frac{\pi_{\mathsf{ref}}(a_-)}{\pi_{\mathsf{ref}}(a_+)} \Big( \frac{C_5 \max\{c_0, 2C_4\} \alpha_T r_{\mathsf{max}}}{N_T(a_+, a_-)} \Big)^{1/\beta} + 2 C_5 \sqrt{T \log T}.$$

Here step (i) follows from the definition of $t_0$ (cf. (C.9)), while step (ii) uses the above two bounds and Lemma 3 with $t_1 = t_0 - 1$ and $t_2 = T$. This immediately implies that

$$N_T(a_+, a_-) \leq C_7 \left( T \frac{\pi_{\mathsf{ref}}(a_-)}{\pi_{\mathsf{ref}}(a_+)} \right)^{\frac{\beta}{\beta+1}} (\alpha_T r_{\mathsf{max}})^{\frac{1}{\beta+1}} + C_7 \sqrt{T \log T}$$

for some sufficiently large constant $C_7 > 0$. This leads to

$$\zeta(a_+, a_-) \leq N_T(a_+, a_-) r_{\mathsf{max}} \leq C_7 \left( T \frac{\pi_{\mathsf{ref}}(a_-)}{\pi_{\mathsf{ref}}(a_+)} \right)^{\frac{\beta}{\beta+1}} \alpha_T^{\frac{1}{\beta+1}} r_{\mathsf{max}}^{\frac{\beta+2}{\beta+1}} + C_7 \sqrt{T \log T} r_{\mathsf{max}}. \tag{C.12}$$

**Case 2.** When $N_T(a_+, a_-) > c_0 \exp(r^\star(a_+) - r^\star(a_-)) \alpha_T r_{\mathsf{max}}$, we have

$$\exp(r^\star(a_+) - r^\star(a_-)) \alpha_T r_{\mathsf{max}} \leq \frac{1}{c_0} N_T(a_+, a_-) \overset{(i)}{\leq} \frac{2}{c_0} [N_T(a_+, a_-) - N_{t_0-1}(a_+, a_-)]$$

$$\overset{(ii)}{\leq} \frac{2 C_5^{1/\beta}}{c_0} T \frac{\pi_{\mathsf{ref}}(a_-)}{\pi_{\mathsf{ref}}(a_+)} \left[ \sigma(r^\star(a_-) - r^\star(a_+))^{1/\beta} + \Big( \frac{2 C_4 \alpha_T r_{\mathsf{max}}}{N_T(a_+, a_-)} \Big)^{1/\beta} \right] + \frac{2 C_5}{c_0} \sqrt{T \log T}$$

$$\overset{(iii)}{\leq} \frac{4}{c_0} \max\{C_5, 2 C_4 C_5/c_0\}^{1/\beta} T \frac{\pi_{\mathsf{ref}}(a_-)}{\pi_{\mathsf{ref}}(a_+)} \exp(r^\star(a_-) - r^\star(a_+))^{1/\beta} + \frac{2 C_5}{c_0} \sqrt{T \log T}.$$

Here step (i) follows from the definition of $t_0$ (cf. (C.9)); step (ii) utilizes Lemma 3 with $t_1 = t_0 - 1$ and $t_2 = T$, as well as (C.11); step (iii) holds since $\sigma(r^\star(a_-) - r^\star(a_+)) \leq \exp(r^\star(a_-) - r^\star(a_+))$ and

$$\frac{2 C_4 \alpha_T r_{\mathsf{max}}}{N_T(a_+, a_-)} \leq \frac{2 C_4 \alpha_T r_{\mathsf{max}}}{c_0 \exp(r^\star(a_+) - r^\star(a_-)) \alpha_T r_{\mathsf{max}}} \leq \frac{2 C_4}{c_0} \exp(r^\star(a_-) - r^\star(a_+)).$$

This immediately implies that for some sufficiently large constant $C_8 > 0$, we have

$$\frac{\pi_{\mathsf{HF}}(a_+)}{\pi_{\mathsf{HF}}(a_-)} = \exp(r^\star(a_+) - r^\star(a_-)) \leq C_8 \Big( \frac{\pi_{\mathsf{ref}}(a_-) T}{\pi_{\mathsf{ref}}(a_+) \alpha_T r_{\mathsf{max}}} \Big)^{\frac{\beta}{\beta+1}} + C_8 \frac{\sqrt{T \log T}}{\alpha_T r_{\mathsf{max}}}. \tag{C.13}$$

Similar to (C.1), we can show that

$$\mathsf{KL}\big(\sigma(r^\star(a_+) - r^\star(a_-))\|\sigma(r(a_+) - r(a_-))\big) = \mathsf{KL}\big(\sigma(r^\star(a_-) - r^\star(a_+))\|\sigma(r(a_-) - r(a_+))\big)$$

$$\overset{(a)}{\geq} \frac{1}{16 r_{\max}}\sigma(r^\star(a_-) - r^\star(a_+))[1 - \sigma(r^\star(a_-) - r^\star(a_+))][r(a_+) - r(a_-) - r^\star(a_+) + r^\star(a_-)]^2$$

$$\overset{(b)}{\geq} \frac{1}{64 r_{\max}}\exp(r^\star(a_-) - r^\star(a_+))[r(a_+) - r(a_-) - r^\star(a_+) + r^\star(a_-)]^2.$$

Here step (a) follows from Lemma 5; step (b) makes use of the fact that $r^\star(a_-) \leq r^\star(a_+)$. Hence we have

$$[r(a_+) - r(a_-) - r^\star(a_+) + r^\star(a_-)]^2$$

$$\leq 64 r_{\max}\frac{\pi_{\mathsf{HF}}(a_+)}{\pi_{\mathsf{HF}}(a_-)}\mathsf{KL}\big(\sigma(r^\star(a_+) - r^\star(a_-)) \| \sigma(r(a_+) - r(a_-))\big). \qquad (\text{C.14})$$

In addition, we have

$$\sum_{i=1}^{t}\mathsf{KL}\big(\sigma(r^\star(a_1^i) - r^\star(a_2^i)) \| \sigma(r^{(t)}(a_1^i) - r^{(t)}(a_2^i))\big) \overset{(i)}{\leq} -2\big[\ell(r^{(t)}, \mathcal{D}^{(t)}) - \ell(r^\star, \mathcal{D}^{(t)})\big] + 2C_3 A r_{\max}\log T$$

$$\overset{(ii)}{\leq} 2\alpha_t\gamma_t + 2C_3 A r_{\max}\log T$$

Here step (i) follows from Lemma 2, while step (ii) utilizes (5.6) and the definition of $\gamma_t$ (cf. (5.2)). This immediately implies that

$$\mathsf{KL}\big(\sigma(r^\star(a_+) - r^\star(a_-)) \| \sigma(r^{(t)}(a_+) - r^{(t)}(a_-))\big) \leq \frac{2\alpha_t\gamma_t + 2C_3 A r_{\max}\log T}{N_t(a_+, a_-)}. \qquad (\text{C.15})$$

Therefore for any $1 \leq n_1 < n_2 \leq N_T(a_+, a_-)$, we have

$$\frac{1}{n_2 - n_1}\bigg(\sum_{n=n_1}^{n_2}\big|r^{(t_n)}(a_+) - r^{(t_n)}(a_-) - r^\star(a_+) + r^\star(a_-)\big|\bigg)^2$$

$$\overset{(i)}{\leq} \sum_{n=n_1}^{n_2}[r^{(t_n)}(a_+) - r^{(t_n)}(a_-) - r^\star(a_+) + r^\star(a_-)]^2$$

$$\overset{(ii)}{\leq} 64 r_{\max}\frac{\pi_{\mathsf{HF}}(a_+)}{\pi_{\mathsf{HF}}(a_-)}\sum_{n=n_1}^{n_2}\mathsf{KL}\big(\sigma(r^\star(a_+) - r^\star(a_-)) \| \sigma(r^{(t_n)}(a_+) - r^{(t_n)}(a_-))\big)$$

$$\overset{(iii)}{\leq} \frac{128 r_{\max}\alpha_T}{n_1}\frac{\pi_{\mathsf{HF}}(a_+)}{\pi_{\mathsf{HF}}(a_-)}\sum_{n=n_1}^{n_2}\gamma_{t_n} + 128 C_3 A r_{\max}^2\log T\frac{n_2 - n_1}{n_1}\frac{\pi_{\mathsf{HF}}(a_+)}{\pi_{\mathsf{HF}}(a_-)}. \qquad (\text{C.16})$$

Here step (i) uses the Cauchy-Schwarz inequality; step (ii) follows from (C.14); step (iii) utilizes (C.15) and the fact that $\{\alpha_t\}$ is monotonically increasing. Following the same analysis as in (5.3) and (5.4), we know that

$$\sum_{n=n_1}^{n_2}\gamma_{t_n} \leq \sum_{n=n_1}^{n_2}\xi_{t_n} + \sum_{n=n_1}^{n_2}\big|r^{(t_n)}(a_+) - r^{(t_n)}(a_-) - r^\star(a_+) + r^\star(a_-)\big|$$

$$\leq C_1 r_{\max}\sqrt{(n_2 - n_1)\log T} + \sum_{n=n_1}^{n_2}\big|r^{(t_n)}(a_+) - r^{(t_n)}(a_-) - r^\star(a_+) + r^\star(a_-)\big|. \qquad (\text{C.17})$$

Taking (C.16) and (C.17) collectively and let $n_2 = 2n_1$, we know that for any $n_1 \leq N_T(a_+, a_-)/2$,

$$\bigg(\sum_{n=n_1}^{2n_1}\big|r^{(t_n)}(a_+) - r^{(t_n)}(a_-) - r^\star(a_+) + r^\star(a_-)\big|\bigg)^2$$

$$\overset{(iii)}{\leq} 128 r_{\max}\alpha_T\frac{\pi_{\mathsf{HF}}(a_+)}{\pi_{\mathsf{HF}}(a_-)}\sum_{n=n_1}^{2n_1}\big|r^{(t_n)}(a_+) - r^{(t_n)}(a_-) - r^\star(a_+) + r^\star(a_-)\big|$$

$$+ 128 r_{\max} \frac{\pi_{\mathsf{HF}}(a_+)}{\pi_{\mathsf{HF}}(a_-)} n_1 \left( \alpha_T C_1 r_{\max} \sqrt{\frac{\log T}{n_1}} + C_3 A r_{\max} \log T \right).$$

This self-bounding relation implies that

$$\sum_{n=n_1}^{2n_1} \left| r^{(t_n)}(a_+) - r^{(t_n)}(a_-) - r^{\star}(a_+) + r^{\star}(a_-) \right| \leq 256 r_{\max} \alpha_T \frac{\pi_{\mathsf{HF}}(a_+)}{\pi_{\mathsf{HF}}(a_-)}$$

$$+ \sqrt{256 r_{\max} \frac{\pi_{\mathsf{HF}}(a_+)}{\pi_{\mathsf{HF}}(a_-)} n_1 \left( \alpha_T C_1 r_{\max} \sqrt{\frac{\log T}{n_1}} + C_3 A r_{\max} \log T \right)}.$$

$$\leq 400 r_{\max} \alpha_T \frac{\pi_{\mathsf{HF}}(a_+)}{\pi_{\mathsf{HF}}(a_-)} + C_1 r_{\max} \sqrt{n_1 \log T} + C_3 n_1 \frac{A r_{\max} \log T}{\alpha_T},$$

where the last relation follows from the AM-GM inequality. By using the above relation recursively, we have

$$\zeta(a_+, a_-) \leq \sum_{k=1}^{\lceil \log T \rceil} \sum_{n=N_T(a_+,a_-)/2^k}^{N_T(a_+,a_-)/2^{k-1}} \left| r^{(t_n)}(a_+) - r^{(t_n)}(a_-) - r^{\star}(a_+) + r^{\star}(a_-) \right|$$

$$\leq C_9 r_{\max} \frac{\pi_{\mathsf{HF}}(a_+)}{\pi_{\mathsf{HF}}(a_-)} \alpha_T \log T + C_9 r_{\max} \sqrt{N_T(a_+, a_-) \log T} + C_9 N_T(a_+, a_-) \frac{A r_{\max} \log T}{\alpha_T}$$

$$\text{(C.18)}$$

for some sufficiently large constant $C_9 > 0$. On the other hand, taking (C.18) and (C.13) collectively yields

$$\zeta(a_+, a_-) \leq C_8 C_9 \left( T \frac{\pi_{\mathsf{ref}}(a_-)}{\pi_{\mathsf{ref}}(a_+)} \right)^{\frac{\beta}{\beta+1}} \alpha_T^{\frac{1}{\beta+1}} r_{\max}^{\frac{1}{\beta+1}} \log T + C_9 r_{\max} \sqrt{N_T(a_+, a_-) \log T}$$

$$+ C_9 N_T(a_+, a_-) \frac{A r_{\max} \log T}{\alpha_T}. \qquad \text{(C.19)}$$

By putting (C.10), (C.12), (C.18) and (C.19) together, we have

$$\zeta(a_+, a_-) \leq C_6 (r_{\max} + \log T) \min \left\{ \frac{\pi_{\mathsf{HF}}(a_+)}{\pi_{\mathsf{HF}}(a_-)} \alpha_T r_{\max}, \left( T \frac{\pi_{\mathsf{ref}}(a_-)}{\pi_{\mathsf{ref}}(a_+)} \right)^{\frac{\beta}{\beta+1}} \alpha_T^{\frac{1}{\beta+1}} r_{\max}^{\frac{1}{\beta+1}} \right\}$$

$$+ C_6 \left( \frac{A N_T(a_+, a_-) \log T}{\alpha_T} + \sqrt{T \log T} \right) r_{\max}$$

always holds for some universal constant $C_6 > 0$.

# D    PROOF OF PROPOSITION 3

Under Assumption 1, we know that for any action pair $(a_+, a_-)$,

$$\min \left\{ \frac{\pi_{\mathsf{HF}}(a_+)}{\pi_{\mathsf{HF}}(a_-)} \alpha_T r_{\max}, \left( T \frac{\pi_{\mathsf{ref}}(a_-)}{\pi_{\mathsf{ref}}(a_+)} \right)^{\frac{\beta}{\beta+1}} \alpha_T^{\frac{1}{\beta+1}} r_{\max}^{\frac{1}{\beta+1}} \right\} \leq \max \left\{ \tau \alpha_T r_{\max}, (\kappa T)^{\frac{\beta}{\beta+1}} \alpha_T^{\frac{1}{\beta+1}} r_{\max}^{\frac{1}{\beta+1}} \right\}.$$

Therefore we have

$$\mathcal{R}(T) \leq C r_{\max} A^2 \sqrt{T \log T} + C \sum_{t=1}^{T} \frac{A r_{\max} \log T}{\alpha_t} + 2C(r_{\max} + \log T) A^2 \tau \alpha_T r_{\max}$$

$$+ C(r_{\max} + \log T) A^2 (\kappa T)^{\frac{\beta}{\beta+1}} \alpha_T^{\frac{1}{\beta+1}} r_{\max}^{\frac{1}{\beta+1}}.$$

By taking

$$\alpha_t = A \log T + t^{\frac{1}{\beta+2}} \left( \frac{r_{\max}}{\kappa} \right)^{\frac{\beta}{\beta+2}} \left( \frac{\log T}{A(r_{\max} + \log T)} \right)^{\frac{\beta+1}{\beta+2}},$$

we can achieve

$$\mathcal{R}(T) \lesssim (r_{\mathsf{max}} + \log T) A^3 \tau r_{\mathsf{max}} \log T + r_{\mathsf{max}} A^2 \sqrt{T \log T}$$

$$+ (r_{\mathsf{max}} + \log T)^{\frac{\beta+1}{\beta+2}} r_{\mathsf{max}}^{\frac{2}{\beta+2}} \kappa^{\frac{\beta}{\beta+2}} A^{\frac{2\beta+3}{\beta+2}} T^{\frac{\beta+1}{\beta+2}} (\log T)^{\frac{1}{\beta+2}}$$

$$+ (r_{\mathsf{max}} + \log T)^{\frac{1}{\beta+2}} A^{\frac{\beta+3}{\beta+2}} \tau r_{\mathsf{max}}^{\frac{2\beta+2}{\beta+2}} \kappa^{-\frac{\beta}{\beta+2}} (\log T)^{\frac{\beta+1}{\beta+2}} T^{\frac{1}{\beta+2}}$$

$$+ (r_{\mathsf{max}} + \log T) A^{\frac{2\beta+3}{\beta+1}} \kappa^{\frac{\beta}{\beta+1}} (\log T)^{\frac{1}{\beta+1}} r_{\mathsf{max}}^{\frac{1}{\beta+1}} T^{\frac{\beta}{\beta+1}}$$

$$\lesssim \tau A^3 r_{\mathsf{max}}^2 \log^2 T + T^{\frac{\beta+1}{\beta+2}} \kappa^\beta r_{\mathsf{max}}^2 A^3 \tau \log^2 T.$$

# E    ANOTHER ASSUMPTION AND THE REGRET BOUND

As an alternaive to Assumption 1, we can also impose the following assumption to capture the relation between human preference $\pi_{\mathsf{HF}}$ and the reference policy $\pi_{\mathsf{ref}}$.

**Assumption 2.** *There exists some quantity $\mu > 0$ such that, for any action pair $(a_+, a_-)$,*

$$\frac{\pi_{\mathsf{HF}}(a_+)}{\pi_{\mathsf{HF}}(a_-)} \le \mu \frac{\pi_{\mathsf{ref}}(a_+)}{\pi_{\mathsf{ref}}(a_-)}.$$

The quantity $\mu$ measures the deviation of human preference from the reference policy. Under Assumption 2, we have

$$\min \left\{ \frac{\pi_{\mathsf{HF}}(a_+)}{\pi_{\mathsf{HF}}(a_-)} \alpha_T r_{\mathsf{max}}, \left( T \frac{\pi_{\mathsf{ref}}(a_-)}{\pi_{\mathsf{ref}}(a_+)} \right)^{\frac{\beta}{\beta+1}} \alpha_T^{\frac{1}{\beta+1}} r_{\mathsf{max}}^{\frac{1}{\beta+1}} \right\}$$

$$\le \min \left\{ \mu \frac{\pi_{\mathsf{ref}}(a_+)}{\pi_{\mathsf{ref}}(a_-)} \alpha_T r_{\mathsf{max}}, \left( T \frac{\pi_{\mathsf{ref}}(a_-)}{\pi_{\mathsf{ref}}(a_+)} \right)^{\frac{\beta}{\beta+1}} \alpha_T^{\frac{1}{\beta+1}} r_{\mathsf{max}}^{\frac{1}{\beta+1}} \right\}.$$

$$\le (\mu T)^{\frac{\beta}{2\beta+1}} (\alpha_T r_{\mathsf{max}})^{\frac{\beta+1}{2\beta+1}}.$$

Putting the above relation with (4.1), we have

$$\mathcal{R}(T) \lesssim r_{\mathsf{max}} A^2 \sqrt{T \log T} + \sum_{t=1}^{T} \frac{A r_{\mathsf{max}} \log T}{\alpha_t} + A^2 \alpha_T r_{\mathsf{max}}^2$$

$$+ (r_{\mathsf{max}} + \log T) A^2 (\mu T)^{\frac{\beta}{2\beta+1}} (\alpha_T r_{\mathsf{max}})^{\frac{\beta+1}{2\beta+1}}.$$

By taking

$$\alpha_t = A + t^{\frac{\beta+1}{3\beta+2}} \left( \frac{r_{\mathsf{max}}}{\mu} \right)^{\frac{\beta}{3\beta+2}} \left( \frac{\log T}{A(r_{\mathsf{max}} + \log T)} \right)^{\frac{2\beta+1}{3\beta+2}},$$

we have

$$\mathcal{R}(T) \lesssim T^{\frac{2\beta+1}{3\beta+2}} \mu^{\frac{\beta}{3\beta+2}} \mathsf{poly}(A, r_{\mathsf{max}}, \log T).$$

# F    TECHNICAL LEMMAS

**Lemma 5.** *For any $x, \delta \in \mathbb{R}$, we have*

$$\mathsf{KL}(\sigma(x) \| \sigma(x + \delta)) \ge \frac{1}{4} \sigma(x) (1 - \sigma(x)) \min\{|\delta|, \delta^2\}.$$

*Proof.* Let $f_x(t) := \mathsf{KL}(\sigma(x) \| \sigma(x + t))$. We have

$$f_x(t) = \sigma(x) \log \frac{\sigma(x)}{\sigma(x + t)} + (1 - \sigma(x)) \log \frac{1 - \sigma(x)}{1 - \sigma(x + t)}$$

$$= \sigma(x) \log \left( \frac{\sigma(x)}{1 - \sigma(x)} \cdot \frac{1 - \sigma(x + t)}{\sigma(x + t)} \right) + \log \frac{1 - \sigma(x)}{1 - \sigma(x + t)}$$

$$= \log \frac{1 + \exp(x + t)}{1 + \exp(x)} - \sigma(x)t = \log \left(1 + \sigma(x)(e^t - 1)\right) - \sigma(x)t.$$

Then we have

$$f_x'(t) = \frac{\sigma(x)e^t}{1 + \sigma(x)(e^t - 1)} - \sigma(x) = \frac{\sigma(x)\left(1 - \sigma(x)\right)\left(e^t - 1\right)}{1 + \sigma(x)(e^t - 1)}.$$

For any $t > 0$, we can check that

$$f_x'(t) > \sigma(x)\left(1 - \sigma(x)\right)\left(1 - e^{-t}\right) \geq \frac{1}{2}\sigma(x)\left(1 - \sigma(x)\right)\min\left\{t, 1\right\},$$

and for any $t \in (0, 1)$ we have

$$f_x'(t) < \sigma(x)\left(1 - \sigma(x)\right)\left(e^t - 1\right) \leq 2\sigma(x)\left(1 - \sigma(x)\right)t.$$

This immediately implies that for $\delta > 0$,

$$\mathsf{KL}\left(\sigma(x)\|\sigma(x + \delta)\right) = f_x(\delta) - f_x(0) = \int_0^\delta f_x'(t)\mathrm{d}t$$

$$\geq \frac{1}{2}\sigma(x)\left(1 - \sigma(x)\right)\int_0^\delta \min\left\{t, 1\right\}\mathrm{d}t$$

$$\overset{(a)}{\geq} \frac{1}{4}\sigma(x)\left(1 - \sigma(x)\right)\min\{\delta, \delta^2\}.$$

Here step (a) holds since $\int_0^\delta \min\{t, 1\}\mathrm{d}t = \delta^2/2$ for $\delta \leq 1$, and $\int_0^\delta \min\{t, 1\}\mathrm{d}t = \delta - 1/2 \geq \delta/2$ for $\delta > 1$.

For $\delta < 0$, we can use the same argument to show that

$$\mathsf{KL}\left(\sigma(x)\|\sigma(x + \delta)\right) \geq \frac{1}{4}\sigma(x)\left(1 - \sigma(x)\right)\min\{-\delta, \delta^2\}.$$

This completes the proof. $\qquad\square$

The following lemma provides a user-friendly version of Freedman's inequality (the Bernstein inequality for martingale differences) (Freedman, 1975; Tropp, 2011).

**Lemma 6.** *Consider a filtration $\{\mathcal{F}_i\}_{i \geq 0}$ and random variables $\{X_i\}_{i \geq 1}$ obeying*

$$|X_i| \leq R \qquad and \qquad \mathbb{E}[X_i|\mathcal{F}_{i-1}] = 0 \qquad for\ all\ i \geq 1.$$

*Define $W_n = \sum_{i=1}^n \mathbb{E}[X_i^2|\mathcal{F}_{i-1}]$, and suppose that $W_n \leq \sigma^2$ holds deterministically for some given quantity $\sigma > 0$. Then for any positive integer $m \geq 1$, with probability exceeding $1 - \delta$ we have*

$$\left|\sum_{i=1}^n X_i\right| \leq \sqrt{8 \max\left\{W_n, \frac{\sigma^2}{2^m}\right\}\log \frac{2m}{\delta}} + \frac{4}{3}R\log \frac{2m}{\delta}.$$

*Proof.* See Li et al. (2021, Section A). $\qquad\square$

