# OpenReview forum: "Towards Efficient Online Exploration for Reinforcement Learning with Human Feedback"
_ICLR.cc/2026/Conference — ICLR 2026 Conference Withdrawn Submission_

### Official Review · Reviewer_yZZ2 · 2025-10-28

**Soundness:** 4
**Presentation:** 2
**Contribution:** 3
**Rating:** 6
**Confidence:** 4

**Summary:**

The paper studies online RLHF and proposes a new uncertainty-based exploration framework. The paper discovers an interesting phenomenon in existing optimism-based exploration methods, i.e., the uncertainty quantification misaligns with the action-pair selection. To correct it, the paper proposes a new exploration algorithm that uses an adaptive calibration policy and sequentially compares the current policy with the calibration policy to reduce the reward uncertainty. A regret guarantee is presented with rate $T^{\frac{\beta + 1}{\beta+2}}$ where $\beta$ is the KL regularization weight, with polynomial dependence on other quantities.

**Strengths:**

The paper is a solid theoretical paper with sharp insight into algorithm design, which is already interesting and relevant enough, from my understanding, to guarantee a potential acceptance.

1. The paper discovers two shortcomings of traditional exploration in preference-based RL in LLMs. First, the compared actions do not directly align with decreasing the uncertainty compared to the calibration policy, which defines the uncertainty bonus. Second, even when we compare actions from the current policy to the calibration policy, a fixed calibration policy is not enough. Both are rigorously shown from counterexamples and propositions.

2. The proposed algorithm, in principle, solves both problems intuitively. A regret guarantee is given with polynomial dependence on all problem parameters. And in the regime without KL regularization, the rate matches the best known regret.

3. For most parts, the paper is quite well-written.

**Weaknesses:**

There are some weaknesses in this paper despite its solid contribution. I will raise my score if, in general, the inconsistency between the theoretical characterization of regret bound in some cases and the intuition of algorithm development is resolved to some extent.

1. The theoretical results with KL regularization are a bit detached from the storytelling of the algorithm design. It is a bit strange in the sense that (consider the example of $\beta = 1$ and examine the large $T$ regime) the proposed algorithm should work better than traditional RLHF exploration algorithms, from an intuitive understanding, but the regret bound looks otherwise. The new algorithm has $T^{2/3}$ while classic algorithm has $T^{1/2}$. The authors should include a deep discussion on why the exponent $\beta / (\beta + 2)$ appears, and explain carefully why this could or could not be fundamental.

2. I feel the regret defined in the regularized objective function is not natural and does not make sense at several points. They make more sense in terms of studying the convergence of policy, but not the regret. Regret should only count the loss of utility in previous rounds and not other notions such as the KL divergence. Essentially, people want a policy with a large reward. For example, if a policy only uses the best arm $a^*$, it should be a very good policy. But in the paper's definition of regret, this policy would incur a large constant regret in some cases. On the other hand, the regret also cumulatively sums the KL distance difference of policies to the reference policy, which does not translate to any physical meaning. I understand the authors are following a line of work that defines regret as it is in the paper, but I feel there is a way to define the regret more naturally and present the results more cleanly, while still recognizing the effect of KL regularization. For example, one could let the regret be $0$ when the policy value $J(\pi)$ is larger than the value of the optimal policy defined by the regularized problem. I do believe the guarantee in the paper could directly translate to a guarantee in an alternative definition where the regret would directly reflect the loss of utility and the distance to true optimality. This may also be part of the reason why the regret guarantee does not match classic exploration in dependence on $T$.

3. The essential principle behind the adaptive calibration policy and comparison to it for exploration is very closely related to the optimism principle in dueling bandit problems, e.g., RUCB, which also uses a dynamic challenger scheme. A discussion of this literature and comparing the proposed algorithm to them would benefit the paper a lot, especially in explaining the idea and motivation behind the proposed method. That is, there is a systematic flaw in classic exploration in LLM RLHF.

4. The paper should use a representative empirical study (a contextual bandit example as the model of this paper should suffice) to validate a few aspects that are not clear from the theory section. First, the regret guarantee seems to be somewhat inconsistent with the beneficiary described in the algorithm development, so empirical evidence would present a lot of insight into whether the rate is a fundamental block (as suspected by the authors), or is merely something from an analysis bottleneck, but the algorithm itself would still perform better than classic exploration. Second, it would strengthen the impact for practitioners to fix the systematic flaw discovered in this paper in LLM RLHF.

5. The claim in the abstract is overly strong. Various papers study online RLHF, but in different settings where the regret (subject to different definitions) could be polynomial in all parameters. For example, plugging in a dueling bandit algorithm into an MAB setting studied in this paper with $\beta = 0$ would lead to a regret bound that is polynomial in all parameters. I suggest the authors remove this claim.

**Questions:**

See weakness

---

### Official Review · Reviewer_Ncqg · 2025-10-29

**Soundness:** 3
**Presentation:** 3
**Contribution:** 2
**Rating:** 4
**Confidence:** 2

**Summary:**

The paper proposes a new exploration strategy for online RLHF, which samples one completion from the previous policy and one completion from the current policy. This strategy is motivated from the perspective of uncertainty reduction, with the intuition that the reward differences from such completion pairs are more informative for policy improvements. Under a multi-armed bandit setting, this strategy is proved to have polynomial regret.

**Strengths:**

1. The topic being considered, the exploration strategy for online RLHF, is of great importance.
2. The paper provides some insight for why current methods such as VPO are not optimal.

**Weaknesses:**

1. There is no empirical evidence for the efficacy of the proposed strategy.

2. The theoretical analysis is based on a multi-armed bandit setting, which assumes a fixed set of actions. However, in this paper actions correspond to model completions. In this case the action space itself is exponentially large. Since the regret bound involves the size of action space, I wonder if such regret bound is meaningful in practice.

3. Although uncertainty reduction is not a novel idea, this paper does not mention any prior strategies in the area of preference modeling.

**Questions:**

1. Since the proposed strategy involves sampling a completion from previous model $\pi^{(t-1)}$，I wonder how this can be implemented in practice. I think keeping an old copy of model parameters is not viable for large models, and caching completions faces storage issue.

---

### Official Review · Reviewer_Bvs6 · 2025-10-31

**Soundness:** 3
**Presentation:** 3
**Contribution:** 2
**Rating:** 2
**Confidence:** 3

**Summary:**

This paper proposes another variant of online/iterative DPO to improve online RLHF. It also analyses the regret in the bandit setting of some proposed alternatives in the literature (XPO, VPO and SELM) to show, for some of them, that regret might scale unfavorably in some parameters. They propose a new algorithm  (a variant close to XPO, except that the second trajectory is sampled not from the reference policy, but from the previous policy one timestep ago) and show in the bandit setting that this algorithm has a regret which, while scaling less favorably in the number of time steps, scales polynomially in all other parameters (like \beta, the KL strength).

**Strengths:**

The paper is clearly written and gives a good summary of most prior work on online sampling and exploration bonuses. It provides clear statements and proofs in the bandit setting, which are discussed so the reader can understand some higher level picture. It gives a good self contained intro to reward modeling and RLHF.

**Weaknesses:**

A major weakness of this work is a total lack of experiments on LLMs (or of any experiments, for that matter). Of course, this paper could be styled as a pure theoretical contribution, but given it actually proposes a new sampling strategy for RLHF for LLMs, and then does all the theory on bandits, not having any experiments beyond the bandit setting is a real problem.
Minor weaknesses:
1. The paper works with a calibration policy \pi_{\cal} which is not standard in all recent RLHF papers (for instance, XPO doesn't have it). As such, I would have liked more of a discussion on why it is being used in this paper and why other papers managed without it.
2. There are more sampling approaches that are not being discussed, when they should be, like the PILAF paper (Feng et al 25). Can you add and compare (beyond just citing it)?
3. I am actually missing a motivation of why it is important to not scale exponentially in some parameters like \beta. In the end, this is a constant in practical implementations.

**Questions:**

Please see the weaknesses 1.-3. for questions.
Typo line 86: SVT --> SFT

---

### Note · Authors · 2025-11-29

I have read and agree with the venue's withdrawal policy on behalf of myself and my co-authors.